# MicroRNA-mediated attenuation of branched-chain amino acid catabolism promotes ferroptosis in chronic kidney disease

Hisakatsu Sone [1], Tae Jin Lee [2], Byung Rho Lee [1], Dan Heo [1], Sekyung Oh [3] & Sang-Ho Kwon [1]

Chronic kidney disease can develop from kidney injury incident to chemotherapy with cisplatin, which complicates the prognosis of cancer patients. MicroRNAs regulate gene expression by pairing with specific sets of messenger RNAs. Therefore, elucidating direct physical interactions between microRNAs and their target messenger RNAs can help decipher crucial biological processes associated with cisplatin-induced kidney injury. Through intermolecular ligation and transcriptome-wide sequencing, we here identify direct pairs of microRNAs and their target messenger RNAs in the kidney of male mice injured by cisplatin. We find that a group of cisplatin-induced microRNAs can target select messenger RNAs that affect the mitochondrial metabolic pathways in the injured kidney. Specifically, a cisplatin-induced microRNA, miR-429-3p, suppresses the pathway that catabolizes branched-chain amino acids in the proximal tubule, leading to cell death dependent on lipid peroxidation, called ferroptosis. Identification of miRNA-429-3p-mediated ferroptosis stimulation suggests therapeutic potential for modulating the branched-chain amino acid pathway in ameliorating cisplatin-induced kidney injury.

A wide range of physical, chemical, and biological stimuli can disturb the homeostasis of body fluids that the kidney exquisitely maintains. These intrinsic or extrinsic insults frequently injure essential kidney tissues, prompting conserved biochemical mechanisms to repair the impairment[1,2]. While the afflicted kidney may be able to restore normal homeostasis from acute disturbance, lingering unresolved impairments would mount a long-lasting pathological state[1,2]. Such a persistent failure of the kidney, chronic kidney disease (CKD) affects more than one-tenth of adults around the world, often compounded by such complications as diabetes and hypertension[3–5]. The high mortality and morbidity associated with CKD have thus warranted research on the biological processes underlying the development of CKD. In this regard, revealing alterations in gene expression that accompany CKD would help better understand and treat the disease.

The injury instigated by exposure to cisplatin, a chemotherapeutic agent to treat a wide range of cancers, is known to trigger CKD to develop[6,7]. While greatly reducing tumor mass, cisplatin can produce various side effects, including neurotoxicity, cardiotoxicity, and nephrotoxicity[8–10]. Hence, CKD can develop as a critical sequela in those cancer patients who have received cisplatin-based chemotherapy, complicating patients' prognosis. As with other CKD inducers, cisplatin can likely elicit tubulointerstitial fibrosis and inflammatory responses in the injured kidney, which would ultimately proceed to a chronic malfunctioning state[11]. This suggests that cisplatin, as well as other CKD inducers, necessarily adjust the gene expression profiles of disturbed tissue to sustain such a pathological state.

To understand how cisplatin-induced kidney injury modifies gene expression to a perpetually adjusted state, here we investigate microRNA (miRNA), a class of small noncoding RNAs consisting of

[1]Department of Cellular Biology and Anatomy, Medical College of Georgia, Augusta University, Augusta, GA 30912, USA. [2]Center for Biotechnology and Genomic Medicine, Medical College of Georgia, Augusta University, Augusta 30912, USA. [3]Department of Medical Science, Catholic Kwandong University College of Medicine, Incheon 22711, South Korea. ✉e-mail: kkwon@augusta.edu

approximately 22 nucleotides in length[12]. miRNA represses the expression of messenger RNA (mRNA) at the post-transcriptional level through base-pairing interactions with target mRNAs, which would consequently undergo either nucleolytic decay or translational attenuation[12]. In the cytoplasm, an RNA-binding protein of the Argonaute (Ago) family, such as Ago2 assembles the miRNA-induced silencing complex (miRISC) by engaging both miRNA and its cognate target mRNA[12]. Indispensable for target mRNA silencing, the miRNA-mRNA interaction in miRISC has been thoroughly delineated in the biophysical and biochemical aspects[12]. However, the identities of target mRNAs for a given miRNA have been mostly deciphered through inexplicit means such as computational prediction, exogenous expression, and reporter assays[13,14]. Accordingly, revealing definitive pairs of a miRNA and its target mRNAs necessitates an analysis specifically designed for detecting endogenous interactions between the two.

Thus, we here aim to uncover the transcriptome-wide landscape of endogenous miRNA-mRNA interactions in the cisplatin-injured kidney. To this end, the present study implements chimeric-eCLIP-seq, a high-throughput sequencing-based approach to identify miRNA and its target mRNA simultaneously, both of which are associated with Ago2 in miRISC[15-17]. In this method, miRISC-engaged miRNAs and mRNAs are crosslinked to the Ago2 protein with irradiating ultraviolet (UV) light to live cells. The cells are then lysed and successively treated with ribonuclease (RNase) to truncate lengthy mRNAs. The resultant Ago2 complex covalently conjugated to both the miRNA and the RNase-treated trimmed mRNA fragment is then collected by immunoprecipitation with an anti-Ago2 antibody. Afterward, the mRNA fragment and the miRNA are ligated to each other, and the resultant chimeric RNA molecules are subsequently subjected to RNA-seq analysis to discover transcriptome-wide endogenous pairs of miRNAs and their target mRNAs. Reflecting the practical difficulty of performing the UV-irradiation procedure, pulling down Ago2 proteins in vivo, and ligating RNA molecules, however, chimeric-eCLIP-seq studies have been undertaken primarily in cultured cell models thus far.

We present here a benchmark for which the chimeric-eCLIP-seq methodology uncovers an organ-level in vivo landscape of miRNA-mRNA interaction, demonstrating it in the mouse kidney injured by cisplatin. Our approach not only enables the determination of in vivo target mRNA inventory of miRNAs in the chronically injured kidney, as similarly achieved by recent studies on other organ systems[15,17]; but also allows for the identification of specific biological processes that miRNAs regulate in the chronically injured state. We provide evidence that miRNAs expressed in the cisplatin-injured kidney downregulate target mRNAs involved in various carbon-compound metabolism, principally centering on mitochondrial catabolic processes. We find that miRNAs in the injured kidney such as miR-429-3p downregulate the biochemical pathway catabolizing select essential amino acids, valine, leucine, and isoleucine, which are collectively termed branched-chain amino acids (BCAAs)[18,19]. We further demonstrate that blunted BCAA catabolism in this manner can engage iron-dependent lipid peroxidation-mediated cell death, called ferroptosis[20-22], thus illuminating an unprecedented role for miRNA-mediated gene regulation in stimulating ferroptotic cell death through increasing BCAA levels in the injured kidney.

## Results

### A mouse chronic kidney disease model induced by cisplatin
Previous studies have simulated chronic kidney disease (CKD) in mice by repeatedly administering low doses of cisplatin to induce kidney injury[23-25]. Therefore, we exploited such a cisplatin-induced kidney injury mouse model to study CKD, aiming at elucidating CKD-associated alterations in gene expression. Since different doses of cisplatin are known to induce varying renal outcomes and mortality[26], we piloted the optimal dose of cisplatin in our model by injecting

either 7, 8, or 9 mg per kg of cisplatin intraperitoneally into 9-week-old male subject mice four times for the initial three weeks (Fig. 1a). Subsequently, mice were left to develop pathological responses in the kidney for an additional four weeks (Fig. 1a). The subject mice were examined for serum content and gene expression of the kidney at the end of the seven-week period to verify our experimental strategy as a CKD model (Fig. 1a).

To assess the degree of chronic injury state initiated by repeated exposure to cisplatin, we measured the level of blood urine nitrogen (BUN). All three doses of cisplatin significantly increased BUN levels at the end of the seven-week period (Fig. 1b), demonstrating that our experimental approach can compromise kidney function, as expected. However, the highest dose used, 9 mg per kg, induced substantial mortality, as shown by Kaplan-Meyer survival analysis (Fig. 1c). Therefore, we used the sublethal dose of 8 mg per kg in the subsequent analyses, as it could induce a sufficient degree of pathological responses, while avoiding unwanted mortality in the subject animals.

Consistent with the observation of increased BUN levels (Fig. 1b), cisplatin (8 mg/kg) administration correspondingly altered the expression of genes related to tissue damage responses, as revealed by quantitative RT-PCR (RT-qPCR) analysis to measure transcript levels (Fig. 1d). Various genes involved in inflammation (*Cxcl12*, *Cx3cl1*, and *Mif*), apoptosis (*Bax*), ferroptosis (*Gpx4* and *Acsl4*), cell cycle arrest (*Timp2* and *Igfbp7*), and fibrosis (*Col1a1*, *Fn1*, and *Vim*) were significantly upregulated at the transcript level in response to cisplatin (Fig. 1d). Expression of the anti-apoptotic gene *Bcl2* does not appear to vary, suggesting that it might not counterbalance the pro-apoptotic response associated with the elevated expression of *Bax* (Fig. 1d). Likewise, the transcript levels of *Tfrc* (ferroptosis) and *Acta2* (fibrosis) genes did not change in the cisplatin-treated kidney (Fig. 1d). However, the level of α-SMA (α-Smooth muscle actin, the protein product of the *Acta2* gene) was found upregulated in the cisplatin-treated kidney along with other fibrosis markers such as Fibronectin, Vimentin, and Collagen (the protein product of the *Col1A1* gene), as revealed by Western blot analysis (Supplementary Fig. 1), suggesting that fibrotic gene expression might be regulated differentially at the transcript and the protein levels.

### Chimeric-eCLIP-seq identifies Ago2-associated, directly interacting pairs of miRNA and its target mRNA in the cisplatin-injured kidney
To understand how cisplatin alters transcript levels of various tissue damage response-related genes (Fig. 1d), we considered miRNAs, each of which can target a range of mRNAs simultaneously, thereby producing a multitude of effects in gene expression. To uncover the transcriptome-wide landscape of miRNA-mRNA interaction, we employed chimeric-eCLIP-seq, an adaptation of CLIP-seq methodology[16], which had been developed for identifying RNA molecules that directly associate with an RNA-binding protein by covalently crosslinking of the two by irradiating ultraviolet (UV) light to live cells. The chimeric-eCLIP-seq method was purposely devised for the analysis of miRISC-loaded RNAs by performing Ago2 immunoprecipitation and subsequent intermolecular ligation between the resultant co-immunoprecipitated miRNA and mRNA fragments, yielding a hybrid chimeric RNA molecule to be subjected to RNA-seq analysis.

We verified that the chimeric-eCLIP-seq procedure is feasible in the mouse kidney by effectively immunoprecipitating UV-crosslinked Ago2 from both the normal and the cisplatin-injured kidneys (Supplementary Fig. 2a), demonstrating that chimeric-eCLIP-seq can analyze the Ago2 complex that is UV-crosslinked in the kidney. Furthering our endeavors, by using control and 8 mg/kg cisplatin-treated mouse kidney samples, we performed total RNA-seq and small RNA-seq analyses in parallel with chimeric-eCLIP-seq to compare profiles of Ago2-engaged miRNA-mRNA pairs with those of expressed mRNAs and miRNAs in the corresponding kidney tissues (Fig. 2a). Total RNA-seq

and small RNA-seq data (Supplementary Data 1 and 2) revealed that the mouse kidney expresses 58.24% and 40.90% of the entirety of the annotated mouse mRNA and miRNA, respectively, reflecting the renal lineage specification of gene expression (Fig. 2b). Analysis of chimeric eCLIP-seq reads of miRNA-mRNA hybrids (Supplementary Data 3) detected 19.68% of mRNA and 35.17% of miRNA when compared to those from our total and small RNA-seq data, respectively (Fig. 2b).

Consistent with previous reports on miRNA-mRNA interaction[27], our chimeric eCLIP-seq data demonstrate that miRNA interacts preferentially with the 3' untranslated region (3'UTR) of mRNA, as revealed by metagene analysis (Fig. 2c and Supplementary Fig. 2b). This positional predilection of miRNA-mRNA interactions on 3'UTR emerged regardless of either omitting or including the procedure of intermolecular ligation between the miRNA and the mRNA fragment co-immunoprecipitated with Ago2. Un-ligated reads (Fig. 2c) display a metagene pattern indistinguishable from that of ligated ones (Supplementary Fig. 2b), indicating that ligation between miRNA and mRNA in performing chimeric-eCLIP-seq did not appreciably distort the Ago2-mediated miRNA-mRNA interaction. Cisplatin treatment did not significantly alter the preference of the miRNA-mRNA interaction for 3'UTR, suggesting that the fundamental molecular characteristics of the miRNA-mRNA interaction are preserved in the kidney injured by cisplatin (Fig. 2c and Supplementary Fig. 2b). Furthermore, although UV crosslinking and intermolecular ligation in chimeric-eCLIP-seq are

inherently inefficient, the miRNA read counts of chimeric-eCLIP-seq and those of the small RNA-seq were significantly correlated with each other in both control and cisplatin samples (Supplementary Fig. 2c), indicating that our chimeric-eCLIP-seq approach did not considerably skew the miRNA profiles during capturing and processing.

The arrangement of miRNAs according to the descending order of the abundance of chimeric reads shows that miR-26a-5p is the most expressed miRNA complexed with Ago2 in the kidney, followed by let-7c-5p, miR-30c-5p, and let-7b-5p (Fig. 2d). Placing the read abundance in this manner also revealed which Ago2-engaged miRNAs can differentially be expressed in response to cisplatin. For example, cisplatin treatment increased and decreased miR-21a-5p and miR-22-3p levels, respectively (Fig. 2d). RT-qPCR analysis with individual miRNAs, such as miR-21a-5p, miR-34a-5p, miR-192-5p, and miR-429-3p confirmed their abundance changes in response to cisplatin treatment (Supplementary Fig. 2d). We also noted that cisplatin induces the expression of miR-429-3p in the Lrp2 mRNA-positive proximal tubules, as revealed by RNAscope in situ hybridization (Supplementary Fig. 2e).

Since pairs of a miRNA and its target mRNA are sequenced concurrently as a ligated, single molecule in chimeric-eCLIP-seq, the detection of a miRNA can instantaneously retrieve a set of directly interacting target mRNAs. We compared the target mRNA sets defined in this manner from our chimeric-eCLIP-seq reads with ones curated in existing miRNA target reference databases such as miRDB[28]. We noted

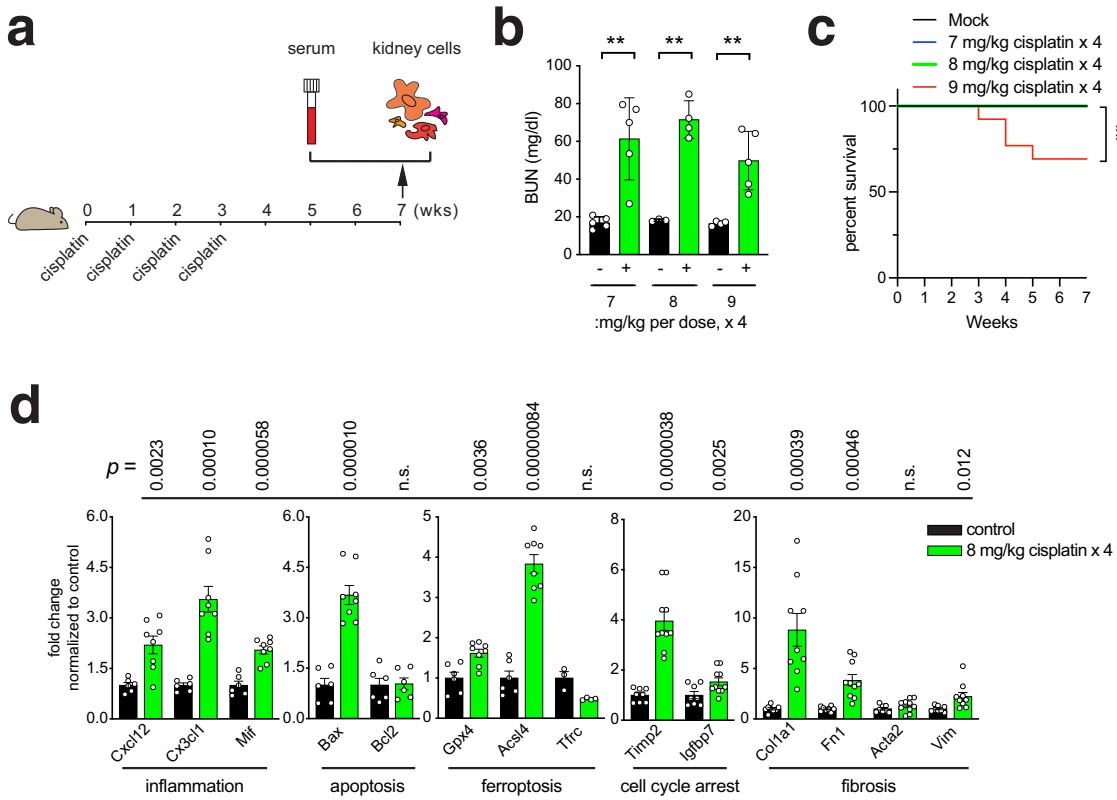

**Fig. 1 | A mouse chronic kidney disease model induced by cisplatin. a** Strategy for inducing chronic kidney disease (CKD) in mice. Varying doses of cisplatin, 7, 8, or 9 mg per kg, were injected intraperitoneally to 9-week-old male mice four times a week as shown. The subject mice were examined for serum content and gene expression of the kidney at the end of the seven-week period. **b** Blood urine nitrogen (BUN) levels in serum measured for the cisplatin-injured mice. Each dot indicates individual, biologically independent mice. $n = 5$, $n = 4$, and $n = 5$ for 7, 8, and 9 mg/kg cisplatin-treated mice, respectively. $n = 5$, $n = 3$, and $n = 4$ for matched mice without cisplatin treatment, respectively. Data are shown as Mean ± SD. **, $p < 0.005$, unpaired, two-sided t-test. **c** Kaplan-Meyer survival analysis for cisplatin-injured mice during the seven-week period. **, $p < 0.005$ as measured by the

Mantel-Cox test. $n = 19$, $n = 9$, $n = 9$, $n = 13$ for mock, 7, 8, and 9 mg/kg cisplatin-treated mice, respectively. **d** RT-qPCR analysis showing the transcript levels of selected genes involved in inflammation, apoptosis, ferroptosis, cell cycle arrest, and fibrosis. Each dot indicates individual, biologically independent mice. Cxcl12, Cx3cl1, Mif, Bax, Bcl2, Gpx4, and Acsl4: $n = 6$ (control) and $n = 8$ (8 mg/kg cisplatin). Tfrc: $n = 3$ (control) and $n = 4$ (8 mg/kg cisplatin). Timp2, and Igfbp7: $n = 8$ (control) and $n = 10$ (8 mg/kg cisplatin). Col1a1: $n = 8$ (control) and $n = 9$ (8 kg/mg cisplatin). Fn1, Acta2, and Vim: $n = 8$ (control) and $n = 10$ (8 kg/mg cisplatin). Data were normalized to their respective control and presented as Mean ± SEM. n.s., not significant ($p > 0.05$), unpaired, two-sided t-test. Source data are provided as a Source Data file.

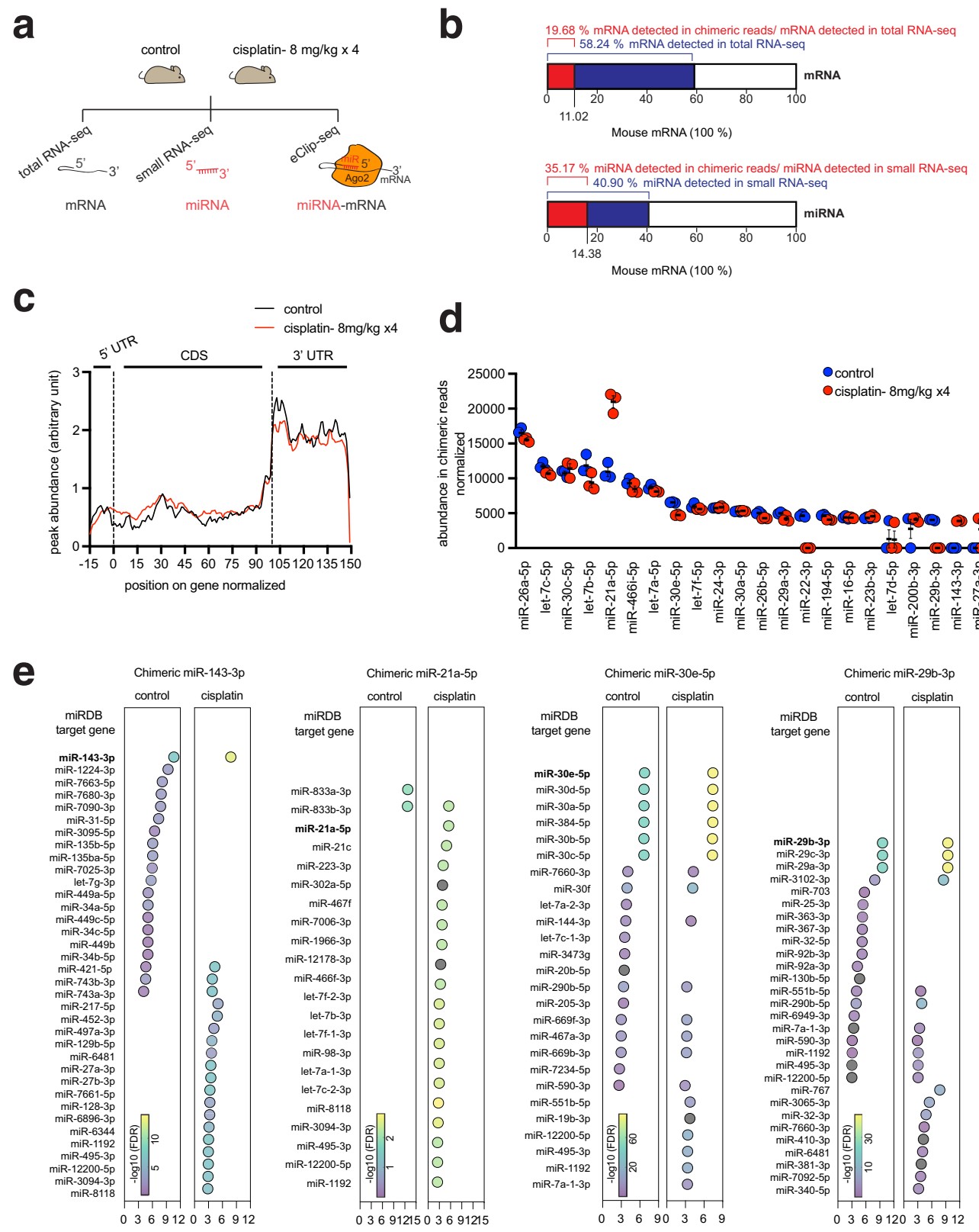

that our set of miR-143-3p chimeric-eCLIP-seq mRNA targets highly enriches constituents of the matching miRDB target set of miR-143-3p (Fig. 2e), supporting that intermolecular ligation events to generate hybrid reads of directly interacting miRNAs and mRNAs were enriched for real interactions rather than background artifacts. Likewise, chimeric-eCLIP-seq target sets of miR-21a-5p, miR-30e-5p, and miR-

29b-3p significantly enrich elements of the corresponding miRDB target sets (Fig. 2e). The high degree overlaps between our target sets and miRDB sets suggest that the chimeric miRNA-mRNA reads were specifically formed through intermolecular ligation within the Ago2-containing miRISC, indicating that our chimeric-eCLIP-seq approach confidently recovered direct mRNA targets of miRNAs.

**Fig. 2 | chimeric-eCLIP-seq analysis identifies Ago2-associated, directly interacting pairs of miRNA and its target mRNA in the cisplatin-injured mouse kidney. a** Approach to determine directly interacting pairs of miRNA and its target mRNA (chimeric-eCLIP-seq) in the cisplatin-injured kidney, as well as to identify mRNAs (total RNA-seq) and miRNAs (small RNA-seq) expressed in matching kidney tissues. **b** Percentages of sequencing reads from total RNA-seq and small RNA-seq (blue) in the entirety of annotated mouse RNA. Percentages of sequencing reads from chimeric-eCLIP-seq reads are shown relative to the total kidney-expressed mRNAs and miRNAs (red). **c** The position of the miRNA reads mapped along the

metagene of mRNA from chimeric-eCLIP-seq performed without UV crosslinking. 5′UTR, 5′ untranslated region. CDS, coding sequence. 3′UTR, 3′ untranslated region. **d** Display of miRNAs ranked by the abundance of miRNA-mRNA chimeric reads of the chimeric-eCLIP-seq. Measurements were made from biologically independent mice. $n = 3$ per group. Data are presented as Mean ± SEM. **e** Plots depicting overlaps between the chimeric-eCLIP-seq dataset of a given miRNA (miR-143-3p, miR-21a-5p, miR-30e-5p or miR-29b-3p) and miRDB target sets, which are ranked by fold enrichment. FDR, false discovery rate. Source data are provided as a Source Data file.

The comparison of our datasets with miRDB target sets further shows that target mRNAs of a particular miRNA can also be targeted by other miRNAs. For example, miR-143-3p targets identified by chimeric-eCLIP-seq are considerably represented in the miRDB target sets of miR-1224-3p, miR-7663-5p, and miR-7680-3p in control; those of miR217-5p, miR452-3p, and miR-497a-3p in samples treated with cisplatin; and numerous other miRDB target sets unique to either treatment (Fig. 2e). Furthermore, the miR-143-3p target set identified by chimeric-eCLIP-seq enriches the constituents of miRDB target sets of miR-421-5p, miR-743b-3p, and miR-743a-3p in both control and cisplatin samples (Fig. 2e). Other target mRNA sets defined by chimeric-eCLIP-seq, such as those of miR-21a-5p, miR-30e-5p, and miR-29b-3p, also display similar characteristics of target assemblage, which can be recovered from multiple miRDB target sets (Fig. 2e). These findings collectively demonstrate that multiple miRNAs can regulate a shared group of mRNAs and that this regulation can be adjusted in response to cisplatin.

### miRNAs expressed in the cisplatin-injured kidney primarily target metabolic pathways

To reveal how cisplatin regulates miRNAs to alter mRNA profiles, we ranked miRNAs by the descending order of the size of their target mRNA sets, considering exclusively those miRNAs and mRNAs that vary in expression in response to cisplatin. This analysis identifies that the miRNA let-7a-5p targets the largest collection of cisplatin-responsive (DE: differentially expressed) mRNAs, followed by miR-30e-5p, miR-24-3p, and miR-27a-3p (Supplementary Fig. 3a), demonstrating that individual miRNAs regulate greatly varying ranges of targets.

Next, we inquired as to how effectively a paired miRNA-mRNA interaction can affect the expression level of the engaged mRNA. Examining how widely target mRNA expressions can vary by 19 topmost cisplatin-induced miRNAs, we noted that for 11 of the 19 miRNAs their targets showed significantly reduced expression upon cisplatin treatment compared with those of nontarget mRNAs, as revealed by cumulative frequency plotting (Fig. 3a). These miRNAs include miR-429-3p, miR-21a-5p, let-7i-5p, let-7b-5p, miR-192-5p, miR-143-3p, miR-34a-5p, miR-199b-3p, miR-181d-5p, miR-130a-3p, and miR-17-5p (Fig. 3a). On the other hand, the remaining eight miRNAs, miR-24-3p, miR-20a-5p, miR-23a-5p, miR-125a-5p, miR-93-5p, miR-27b-3p, miR-27a-3p, and miR-200c-3p, did not significantly alter the expression levels of their target mRNAs relative to those of nontarget ones (Fig. 3a). These results indicate that select cisplatin-induced miRNAs can cause their target mRNA expressions to wane.

Notably, the direction of the changes in target mRNA expression in response to cisplatin can indicate the biological processes that target mRNAs of a particular miRNA would participate in, as revealed by pathway analysis using the KEGG database (Supplementary Fig. 3b)[29,30]. The mRNA targets that were upregulated in response to cisplatin highly represent such gene ontology (GO) terms as glutathione metabolism, fatty acid metabolism, and Cytochrome P450-drug metabolism, among others (Supplementary Fig. 3; Group 1). These GO terms for upregulated genes appear to center principally on anti-oxidation processes (Supplementary Fig. 3b; Group 1). On the other hand, the downregulated target mRNAs constitute such GO terms as

TCA cycle (the citrate cycle), BCAA catabolism, oxidative phosphorylation, propanoate metabolism, and β-alanine metabolism as being the most significantly enriched ones (Supplementary Fig. 3b; Group 2). Mostly related to mitochondrial catabolic pathways, these GO terms associated with downregulated target mRNAs (Supplementary Fig. 3b; Group 2) share a core set of genes such as *Abat*, *Aldh6a1*, and *Ehhadh*, expressions of which decreased concomitantly in response to cisplatin (Supplementary Fig. 3c).

Examination of the downregulated mRNAs targeted by 11 cisplatin-induced miRNAs (Fig. 3a) continually finds that metabolism-related KEGG GO terms are enriched in these target mRNAs (Fig. 3b). These GO terms range from carbon metabolism and citrate cycle (TCA cycle) to butanoate metabolism and retinol metabolism (Fig. 3b). Downregulated target mRNAs are also involved in various GO terms for cell signaling (Fig. 3b). Of the metabolic pathways suggested by these downregulated target mRNAs, catabolic processes of various amino acids and other carbon compounds recurrently emerge, including degradation of valine, leucine, and isoleucine, which are collectively referred to as branched-chain amino acids (BCAAs)[18,19].

When cisplatin-induced miRNAs, let-7b-5p, miR-21a-5p, miR-34a-5p, miR-192-5p, miR-429-3p, miR-143-3p, and let-7i-5p, were individually analyzed, profiles of their mRNA targets further corroborated their implication in metabolic pathways. While also enriching non-metabolic pathways with 73.07% of genes, the target mRNAs of these selected miRNAs feature various metabolic pathways as a coherent theme with such a large share of 26.93% of genes (Fig. 3c). Principally being catabolic processes, these pathways include citrate cycle (TCA), pyruvate metabolism (PYR), carbon metabolism (C), glycolysis (GLY), propanoate degradation (PRO) and BCAA catabolism (BAC), most of which constituents appear to decrease in expression by regulator miRNAs (Fig. 3c). All these results suggest that cisplatin-induced miRNAs can modify the transcriptomic landscape by downregulating their target mRNAs to alter cellular metabolisms, such as catabolic pathways of select amino acids and other carbon compounds.

### The cisplatin-induced miRNA miR-429-3p regulates ferroptosis gene expression in the kidney

To appreciate the biological implications that the specific target mRNA collection of a miRNA has in the injured kidney, we sought to deplete a cisplatin-induced miRNA. To this end, we injected an LNA inhibitor for miR-429-3p, one of the 11 cisplatin-induced miRNAs that downregulate their target mRNA expression (Fig. 3a), into the tail vein of subject mice together with intraperitoneally administered cisplatin once a week four times and waited an additional four weeks (Fig. 4a). We confirm that the LNA inhibitor greatly reduced miR-429-3p expression to an undetectable level in cisplatin-damaged kidney, as measured by RT-qPCR analysis (Fig. 4b).

The mRNA transcriptome of the cisplatin-injured kidney distinctly responded to the LNA-mediated inhibition of miR-429-3p, as assessed by total RNA-seq analysis (Supplementary Data 4). Principal component analysis revealed that the inhibitor-treated kidney had developed mRNA expression profiles that were qualitatively dissimilar from those of the untreated kidney (Fig. 4c). Furthermore, LNA-mediated inhibition appears to have interrupted miR-429-3p-mediated downregulation of target mRNAs. The expressions of the miR-429-3p target

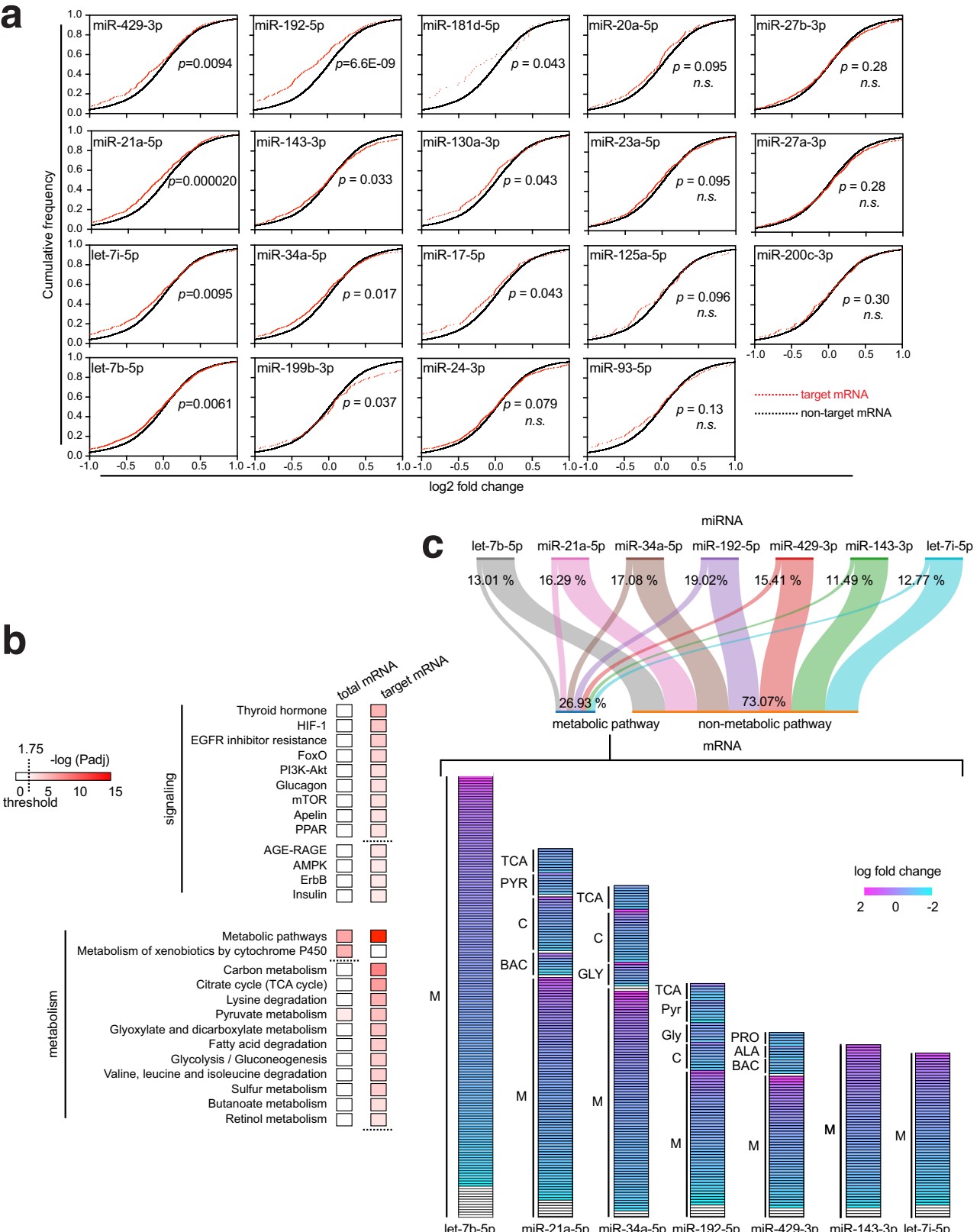

mRNAs identified from the chimeric-eCLIP-seq were found to increase by inhibition mediated by the LNA inhibitor, compared with those of the nontarget mRNAs, as shown in the cumulative frequency plot (Fig. 4d). This finding demonstrates that the LNA inhibitor can effectively interfere with the normal repression of target mRNA expression

by miR-429-3p, while confirming the target mRNAs identified from the chimeric-eCLIP-seq.

Complimentary RT-qPCR analysis of individual genes assured the LNA-induced transcriptomic profile alterations. When analyzed for a set of tissue damage-related genes, the inhibitor-treated kidneys

**Fig. 3 | miRNAs expressed in the cisplatin-injured kidney principally target metabolic pathways. a** Cumulative frequency plots for fold changes of target mRNA expression (red dotted lines) for each of 19 top-most cisplatin-induced miRNAs compared with non-target mRNAs (black dotted lines). The statistical significance of the distribution between non-target mRNA expression and target mRNA expression was determined using the Kolmogorov-Smirnov test (one-sided). *n.s.* not significant ($p > 0.05$). *p* values are FDR-adjusted. **b** KEGG pathway analysis for target mRNAs of 11 miRNAs, of which mRNA targets show significant expression reduction (Fig. 3a). Each gene ontology (GO) term is represented for significance of

mRNA enrichment with a color code with FDR-adjusted *p* value (Padj) from one-sided hypergeometric test, ShinyGO. **c.** KEGG pathway analysis for target mRNAs of 7 selected cisplatin-induced miRNAs. A total of 26.93% of target mRNAs constitutes metabolic pathway GO terms. Fold expression changes of each target mRNA with metabolic pathway GO terms are represented as individual bars with color-coded fold change. TCA citrate cycle, PYR pyruvate metabolism, C carbon metabolism, GLY glycolysis, PRO degradation of propanoate, ALA β-alanine metabolism, BAC branched-chain amino acid (BCAA) catabolism. Source data are provided as a Source Data file.

exhibited reduced expressions of ferroptosis-related genes, such as *Acsl4* and *Tfrc* but not those of genes involved in inflammation, apoptosis, cell cycle arrest, and fibrosis (Fig. 4e). We further noted that inhibition of miR-429-3p can differentially regulate the expressions of ferroptosis-related genes, decreasing those of *Acsl4* and *Tfrc*, but not *Gpx4* (Fig. 4e). These results suggest that miR-429-3p preferentially controls the Acsl4 and Tfrc branches of the ferroptosis pathway in the cisplatin-injured kidney.

To assess the functional consequences of reduced Acsl4 and Tfrc expression in injured kidneys treated with the miR-429-3p LNA inhibitor, we examined cellular lipid peroxidation, a pivotal event in the ferroptosis process. In response to cisplatin, we observed an increase in the lipid peroxidation marker, 4-hydroxy-2-nonenal (4-HNE)[31, 32], within LRP2-positive proximal tubule kidney cells, as demonstrated by immunofluorescence microscopy (Fig. 4f). This observation is consistent with the up-regulation of *Ascl4* expression induced by cisplatin. However, upon administration of the miR-429-3p LNA inhibitor, there was a remarkable reduction in the accumulation of 4-HNE in proximal tubule cells damaged by cisplatin (Fig. 4g). These findings suggest that miR-429-3p promotes ferroptosis by modulating the expression of key components within the ferroptosis pathway, including Acsl4 and Tfrc.

## Stimulation of branched-chain amino acid catabolism can blunt ferroptosis

The GO term compiled as branched-chain amino acid (BCAA) catabolism (BAC) takes a significant share of metabolism-related genes that miRNAs, miR-429-3p and miR-21a-5p, can repress in the kidney injured by cisplatin (Fig. 3c). In the BCAA catabolic pathway, branched-chain α-ketoacid dehydrogenase kinase (BCKDK) is known to phosphorylate and inactivate branched-chain α-ketoacid dehydrogenase (BCKDH), the rate-limiting enzyme that propels the BCAA catabolic pathway[18]. A small molecule called BT2 (3,6-dichlorobenzo(b)thiopene-2-carboxylic acid) was developed to inhibit the catabolic activity of BCKDK, thereby facilitating the BCAA catabolic pathway through liberating BCKDH from the BCKDK suppression[33]. We therefore explored the possibility that BT2-mediated boosting of BCAA catabolism produces an effect akin to that of the LNA-mediated inhibition of miR-429-3p.

Thus, while administering cisplatin in the seven-week CKD induction period, we also injected BT2 intraperitoneally to subject mice eight times twice a week during the initial four weeks (Fig. 5a). Examination of gene expression by RT-qPCR revealed that most of the tissue damage-related genes tested did not appreciably respond to BT2. The transcript levels examined for inflammation, apoptosis, cell cycle arrest, and fibrosis fluctuated little (Fig. 5a). On the other hand, the transcript levels of ferroptosis pathway genes, such as *Acsl4* and *Tfrc* markedly decreased, while that of another ferroptosis gene *Gpx4* did not (Fig. 5a). Reminiscent of the selective downregulation of the ferroptosis genes *Acsl4* and *Tfrc* by LNA-mediated inhibition of miR-429-3p (Fig. 4e), this observation indicates that BT2-mediated stimulation of the BCAA catabolic pathway can blunt the Acsl4 and Tfrc branches of the ferroptosis pathway.

Substantiating the association between BCAA catabolism and ferroptosis, BT2 greatly reduced the protein level of Acsl4 in the cisplatin-injured kidney tissue, which would otherwise express Acsl4

protein at an elevated level, as revealed by immunofluorescence analysis (Fig. 5b, top). Notably, BT2 specifically caused the kidney proximal tubule, marked by LTL, to diminish the expression of Acsl4, which instead persisted exclusively outside the proximal tubule. This observation demonstrates that BT2 can selectively target the proximal tubule in the kidney to decrease Acsl4 expression. Importantly, the reduction in Ascl4 expression appears to mitigate ferroptosis, as the administration of BT2 effectively countered the otherwise elevated expression of the lipid peroxidation marker 4-HNE in the kidney injured by cisplatin (Fig. 5b, bottom).

The BT2-induced Acsl4 repression was recapitulated in BUMPT cells, a cell line of the mouse kidney proximal tubule origin. That is, BT2-treated BUMPT cells expressed significantly reduced levels of Acsl4 protein in response to either 1 or 2 μM cisplatin but not considerably to a higher 10 μM cisplatin, as revealed by quantitative analysis of Western blotting (Fig. 5c and Supplementary Fig. 4).

The BT2-induced downregulation of Acsl4 suggests that enhancing BCAA catabolism could consequently blunt Acsl4-mediated ferroptosis engagement. To test this possibility, we performed Alamar Blue assay, which is devised to measure cell survival of ferroptotic cell death. The addition of BT2 to the culture of BUMPT cells indeed rescued cells from cisplatin-induced ferroptosis (Fig. 5d), implying a crucial protective role for BT2 in thwarting ferroptosis by enhancing BCAA catabolism.

Our chimeric eClip-seq data have revealed that multiple miRNAs, including miR-429-3p and miR-21a-5p, can interact with mRNAs encoding components of the BCAA catabolism pathway (Fig. 3c). To investigate the extent of miR-429-3p in the induction of BCAA catabolism-associated ferroptosis, we conducted experiments using ex vivo mouse kidney cells. For this purpose, cells dissociated from the mouse kidney were depleted of miR-429-3p using an LNA inhibitor for 48 hours and subsequently treated with cisplatin in the presence or absence of BT2 for an additional 30 hours (Supplementary Fig. 5a). The majority of these isolated kidney cells, as indicated by the expression of the kidney proximal tubule marker LRP2, exhibited efficient depletion of miR-429-3p expression when treated with the LNA inhibitor (Supplementary Figs. 5b and c). Interestingly, combined treatment of cells with BT2 and the miR-429-3p LNA inhibitor augmented the reduction both in Ascl4 expression (Supplementary Fig. 5d) and in the level of peroxidized lipid 4-HNE (Supplementary Fig. 5e) in response to cisplatin, compared to single treatment of either BT2 or the LNA inhibitor. These results suggest that additional miRNAs (e.g., miR-21a-5p) can inhibit BCAA catabolism together with miR-429-3p to promote ferroptosis. The additive effect of the combined treatment of BT2 and the miR-429-3p LNA inhibitor further suggests that additional mechanisms, which depend exclusively either on BCAA catabolism or on miR-429-3p but do not operate in the miR-429-3p-induced BCAA catabolic attenuation pathway, can also contribute to cisplatin-induced ferroptosis.

To summarize, our data suggest that cisplatin-induced miRNAs, such as miR-429-3p, repress the expressions of mRNAs encoding key components of the BCAA catabolic pathway, leading to increased intracellular BCAA levels, which subsequently promote ferroptotic cell death.

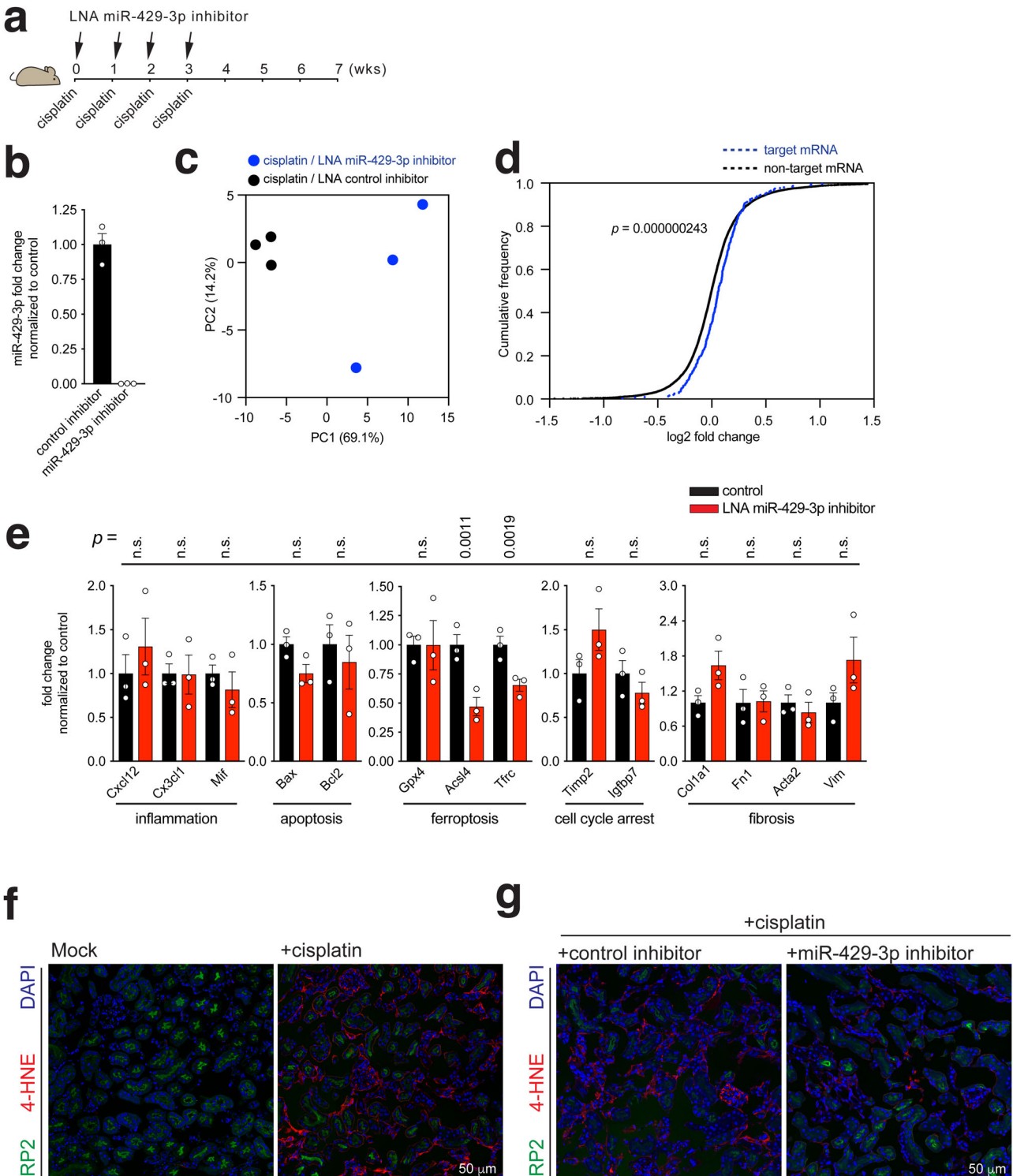

**Fig. 4 | The cisplatin-induced miRNA miR-429-3p regulates ferroptosis gene expression in the kidney. a** Strategy for injecting via tail vein the LNA inhibitor for miR-429-3p while inducing CKD with cisplatin in mice. **b** RT-qPCR analysis for miR-429-3p expression in the presence or absence of the LNA inhibitor. Data were normalized to their control and plotted as Mean ± SEM. Measurements were made from biologically independent mice (*n* = 3 per group). **c** Principal component analysis of mRNA reads of total RNA-seq performed with or without the LNA miR-429-3p inhibitor treatment. **d** Cumulative frequency plot for fold expression changes of target mRNAs for miR-429-3p (blue dotted line) compared with non-target mRNAs (black dotted line). The statistical significance of the distribution between non-target mRNA expression and target mRNA expression was determined using the Kolmogorov-Smirnov test (one-sided). **e** RT-qPCR analysis showing the transcript levels of selected genes involved in inflammation, apoptosis, ferroptosis, cell cycle arrest, and fibrosis. Each dot indicates individual, biologically independent mice (*n* = 3 per group). Data were normalized to their respective control and presented as Mean ± SEM. n.s., not significant (*p* > 0.05), unpaired, two-sided *t*-test. **f** and **g** Immunofluorescence microscopy showing 4-HNE expression in the cisplatin-injured kidney in the presence or absence of miR-429-3p inhibitor with LRP2 as a kidney proximal tubule marker and DAPI as a nucleus marker. Source data are provided as a Source Data file.

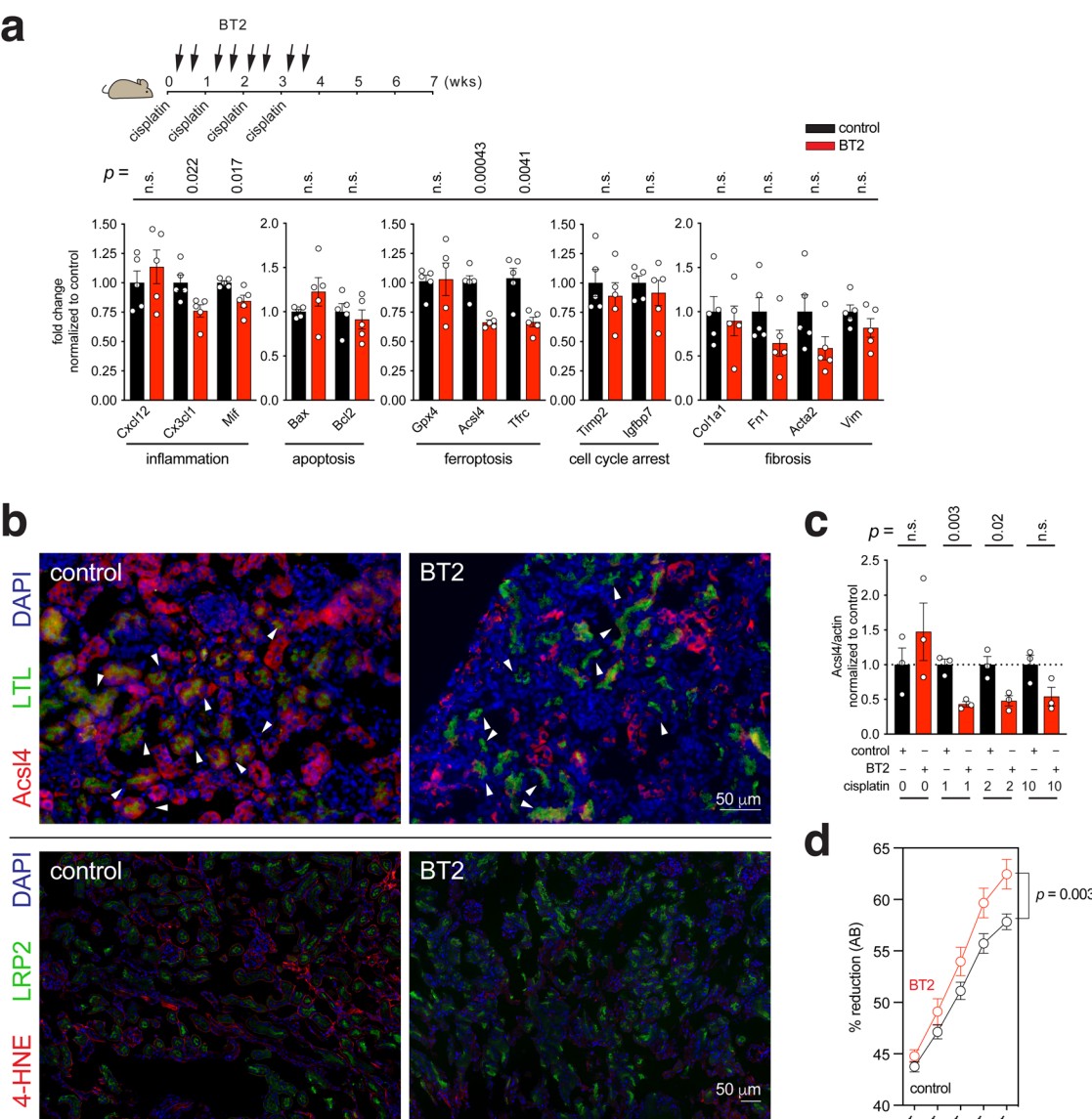

**Fig. 5 | Stimulation of branched-chain amino acid catabolism can blunt fer-roptosis. a** (Top) Strategy for injecting intraperitoneally BT2 while inducing CKD with cisplatin in mice. (Bottom) RT-qPCR analysis showing the transcript levels of selected genes involved in inflammation, apoptosis, ferroptosis, cell cycle arrest, and fibrosis. Each dot indicates individual, biologically independent mice ($n = 5$ per group). Data were normalized to their respective control and presented as Mean ± SEM. n.s., not significant ($p > 0.05$), unpaired, two-sided t-test. **b** Immunofluorescence microscopy showing Acsl4 (Top) and 4-HNE (Bottom) expression in the cisplatin-injured kidney in the presence or absence of BT2 with Lotus tetragolonobus lectin (LTL) and LRP2 as kidney proximal tubule markers and

DAPI as a nucleus marker. **c** Quantification of Acsl4 protein expression in BUMPT cells of the mouse kidney proximal tubule origin from three independent experiments. Each dot represents biologically independent samples ($n = 3$ per group). Data were normalized to their respective control and presented as Mean ± SEM. n.s., not significant ($p > 0.05$), unpaired, two-sided t-test. **d** Alamar Blue assay to measure cell survival of ferroptotic cell death. Each dot represents biologically independent samples ($n = 6$ per group). Data were normalized to their respective control and presented as Mean ± SEM. Statistical significance was determined using a two-way RM ANOVA test. Source data are provided as a Source Data file.

## Discussion

Transcriptome-wide gene expression alterations accompany home-ostasis disturbance to a perpetually malfunctioning state such as CKD[34]. The regulatory macromolecule miRNA can pervasively adjust the transcriptomic activity to elicit responses that are appropriate for the state of the disease. Notwithstanding the recent advent of high-throughput sequencing methodologies that allow scrutiny of tran-scriptomes at multiple levels, the target mRNA inventory for miRNA expressed in a pathological condition such as CKD has not been thoroughly registered. Implementing chimeric-eCLIP-seq in vivo in the mouse kidney injured by cisplatin, we have here unveiled the tran-scriptomic landscape that miRNAs pair with their target mRNAs in cisplatin-induced CKD; and identified biological processes that these

miRNA-mRNA interactions mediate to establish the pathological state of CKD. We find that cisplatin-induced miRNAs, such as miR-429-3p, downregulate those mRNAs involved in various mitochondrial cata-bolic processes, including branched-chain amino acid (BCAA) cata-bolism, illuminating an unprecedented relationship in which miRNA-mediated blunting of BCAA catabolism can lead to ferroptotic cell death in cisplatin-induced CKD.

The present study presents a catalog of in vivo miRNA-mRNA interactions in the context of CKD, while analogous studies on other organs have been reported[15,17]. This feat was achieved by applying the chimeric-eCLIP-seq approach[35,36] to a mouse model of cisplatin-induced CKD. Although the chimeric-eCLIP-seq methodology has high expectations of uncovering direct interactions across the

transcriptome between miRNA and its target mRNA in various contexts, it has been predominantly implemented in cultured cell models. The dearth of in vivo applications of chimeric-eCLIP-seq may be attributed to the difficulty of achieving adequate crosslinking of Ago2 with miRISC-engaged RNAs in tissues and organs with ultraviolet (UV) light, which can penetrate only thin layers of cells. Demonstrating the feasibility of UV crosslinking in the snap-frozen and processed kidney, our chimeric-eCLIP-seq approach, along with comparable studies on other organ systems[15,17], can serve as a benchmark for similar investigations into revealing in vivo miRNA-mRNA interactions in various tissues and organs, many of which might otherwise be regarded incompatible with chimeric-eCLIP-seq thus far.

The chimeric-eCLIP-seq approach has now elucidated that metabolic alteration constitutes a major pillar in miRNA-regulated biological processes in the kidney injured by cisplatin. Various metabolic pathways ranging from the citrate (TCA) cycle and BCAA catabolism to oxidative phosphorylation and propanoate metabolism were specifically targeted and downregulated by miRNAs expressed in the injured kidney (Supplementary Fig. 3b). This implication became more apparent when we analyzed target mRNA sets of 11 of the most effective miRNAs in repressing their targets (Fig. 3a, b). These metabolic pathways consist primarily of mitochondrial catabolic processes, including the citrate cycle, pyruvate metabolism, fatty acid degradation, etc. (Fig. 3b). Notably, the involvement of metabolism appears less evident with the analysis of the total RNA-seq than with that of the chimeric-eCLIP-seq (Fig. 3b). While total RNA-seq identified the GO term of metabolic pathways to some extent as a specific downregulated biological process, it was unable to feature as various detailed metabolism-related GO terms as the chimeric-eCLIP-seq could (Fig. 3b), demonstrating the utility and competence of the chimeric-eCLIP-seq approach in identifying otherwise obscure biological processes in the transcriptomic landscape.

Extending the metabolism implication (Fig. 3c), analysis of individual miRNAs identified a hitherto uncharted miRNA-regulated association between branched-chain amino acid (BCAA) catabolism[18,19] and the iron-dependent lipid peroxidation-dependent cell death called ferroptosis[20–22] (Figs. 4 and 5). The BCAA catabolic (BAC) pathway emerged specifically downregulated by miR-429-3p, as well as miR-21a-5p (Fig. 3c). In particular, LNA-mediated inhibition of miR-429-3p selectively decreased the transcript levels of the fundamental ferroptosis pathway genes, *Acsl4* and *Tfrc*, while not affecting those involved in inflammation, apoptosis, cell cycle arrest, and fibrosis (Fig. 4e), Furthermore, blunting of the BCAA catabolic pathway with the small molecule BT2 decreased the transcript levels of *Acsl4* and *Tfrc* (Fig. 5a), analogous to the effect of LNA-mediated inhibition of miR-429-3p (Fig. 4e). Combined, these observations suggest that miR-429-3p can target and suppress those mRNAs that help catabolize cellular BCAAs and that the consequential increase in BCAA levels can facilitate ferroptosis by upregulating *Acsl4* and *Tfrc* in the injured kidney.

BT2-mediated reduction in cellular BCAA levels would therefore protect the injured kidney from mounting ferroptosis, as evidenced by the immunofluorescence microscopy for the lipid peroxidation marker 4-HNE, as well as the Alamar Blue assay (Fig. 5b, d). On the contrary, high levels of cellular BCAA would be conducive to instigating ferroptosis as a signature response to kidney injury. To support this notion, a recent study showed that upregulation of BCAA level by the transcriptional effector Krüppel-like factor 6 (Klf6) aggravates the state of kidney injury[37]. Another recent study reported that branched-chain amino acid aminotransferase 2 (BCAT2), which converts BCAA to branched-chain keto acid (BCKA) in the BCAA catabolic pathway, can suppress ferroptosis in cancer cells[38]. Extending this realization of the role for BCAA in inducing ferroptosis, our data unveil a specific, dedicated miRNA-regulated transcriptomic program that relates

blunted BCAA catabolism, thus accumulating intracellular BCAA, to ferroptosis engagement.

Intriguingly, we observed that cisplatin can induce Acsl4 expression and ensuing ferroptosis, as indicated by elevated lipid peroxidation, preferentially in the proximal tubule of the kidney (Figs. 4f, g and 5b). This regional specificity of the cisplatin-induced ferroptotic response appears to be related to the corresponding, kidney proximal tubule-restricted expression of miR-429-3p, as revealed by RNAscope™ in situ hybridization (Supplementary Fig. 2e). Accordingly, we find that BT2 acts predominantly on the proximal tubule of the kidney to reverse the cisplatin-induced ferroptosis (Fig. 5b).

Notably, the kidney proximal tubule cells are thought to rely on fatty acid oxidation to gain most of the energy[39]. Since cisplatin can impair the integrity of mitochondria[11], impeding the citrate cycle and the electron transfer chain, mitochondrial fatty acid oxidation might not provide cisplatin-injured proximal tubule cells with adequate energy. We speculate that metabolites resulting from the enhanced BCAA catabolism in response to BT2 might supply either alternative or additional sources for energy production. Generated by rate-limiting branched-chain α-ketoacid dehydrogenase (BCKDH) and other enzymes in the catabolic pathway of BCAA, various conjugates of coenzyme A (CoA), such as isovaleryl-CoA, 2-methylbutyl-CoA and isobutyryl-CoA, and propionyl-CoA are known to feed into the citrate cycle[40]. Therefore, BT2 may facilitate the use of these CoA conjugates to maintain or improve the output of the citrate cycle, thus meeting the energy demand of proximal tubule cells injured by cisplatin.

Interfering with BCAA catabolism therefore has great therapeutic potential for ameliorating kidney injury states such as CKD. For example, the reduction of ferroptosis gene expression by the LNA inhibitor of miR-429-3p (Fig. 4e) can be utilized therapeutically to halt unwanted ferroptosis. Our pathway analysis revealed that the BCAA catabolic pathway can be suppressed not only by miR-429-3p but also by miR-21a-5p (Fig. 3c). Indeed, a recent study presented that miR-21a-5p can stimulate fibrosis via suppressing metabolic pathways in the injured kidney[41]. Therefore, we envision that combined LNA-mediated inhibition of both miR-429-3p and miR-21a-5p would help to formulate a better CKD-treating strategy. In addition to the leads, BCAA catabolism agents such as the inhibitors of branched-chain α-ketoacid dehydrogenase kinase (BDKDK), BT2 and sodium phenylbutyrate (NaPB), can also be exploited therapeutically to ameliorate CKD. Notably, since both BT2 and NaPB are also known to improve the pathological condition of insulin resistance[42,43], this therapeutic strategy would also benefit CKD patients with combined complications such as type 2 diabetes. Finally, given the popularity of cisplatin in treating a wide range of cancers, we expect that cisplatin treatment in combination with LNA inhibitors or BCKDK inhibitors could help minimize the occurrence of unwanted nephrotoxicity incidental to cisplatin while decreasing tumor mass.

## Methods
### Animals
All mouse studies were conducted in accordance with the National Institutes of Health guidelines and protocols were approved by the Medical College of Georgia at Augusta University Institutional Care and Use of Laboratory Animals. C57BL/6 male mice from in-house colonies were kept at 20-24 °C and 30-70% humidity with a 12-hour light/12-hour dark cycle with free access to water and standard food. To model the chronic effects of cisplatin-induced nephrotoxicity, 9-week-old male mice matched in age were randomly divided into two groups (sham vs treated). Mice received saline vehicle or cisplatin with three different doses (7, 8, and 9 mg of cisplatin per kg body weight) as shown in Fig. 1a via intraperitoneal injection once a week for four weeks. To inhibit miRNA activity, chemically synthesized phosphorothioate backboned, locked nucleic acid (LNA) miRNA inhibitor

(Qiagen) (Supplementary Table 1) was resuspended in Dulbecco's phosphate buffered saline and filter-sterilized with a 0.2 μm syringe PES filter to make the final concentration of 4 mg/ml. 20 mg of the resuspended inhibitor per kg body weight per injection was administered to mice through tail vein injection as indicated in Fig. 4a.

To test the effect of BT2 administration in cisplatin-induced kidney injury, BT2 (3,6-dichloro-benzo[b]thiophene-2-carboxylic acid, CAS # 34576-94-8) (Supplementary Table 1) was dissolved in dimethylsulfoxide and diluted in warm sodium bicarbonate buffer to obtain final concentrations of 2 mg/ml BT2/ 100 mM sodium bicarbonate buffer (pH 8.6) for delivery. Mice received vehicle or 20 mg BT2 per kg body weight via intraperitoneal injection as shown in the experimental schedule (Fig. 5a). Blood samples were taken from the lateral tail vein. Terminal kidney collection was performed on mice under isoflurane anesthesia with cervical dislocation. Kidneys were either snap-frozen in liquid nitrogen and stored at -80 °C until used or immediately used for downstream applications.

## Cell culture

The immortalized mouse renal proximal tubule cell line, BUMPT (Boston University Mouse Proximal Tubule clone 306) was originally developed by Lieberthal et al.[44] and was cultured as previously described[45]. Mycoplasma contamination of parental BUMPT cells used in this study was tested with LookOut mycoplasma PCR detection kit (Sigma). For ex vivo renal cell culture, Kidneys from male mice aged 5 or 6 weeks underwent digestion using 130 μg/ml of type 2 collagenase (Worthington Biochemical) in DPBS at 37 °C for 5 minutes. The digestion was performed using a gentle MACS dissociator (Miltenyi Biotec) with programming Multi-E-01 and then B-01. The resulting mixture was diluted with renal epithelial cell growth medium (REGM, Lonza) and passed through a 100-μm cell strainer (Fisher). The filtrate that passed through the strainer was subsequently centrifuged at 70× $g$ for 5 minutes. The resulting cell pellet was resuspended in REGM and underwent another centrifugation step at 70× $g$ for 5 minutes. The collected cell pellet was plated on Nunclon Delta surface-treated culture dishes and cultured in REGM, supplemented with 0.5% fetal bovine serum, 10 ng/ml epidermal growth factor, 35 ng/ml hydrocortisone, 0.5 μg/ml epinephrine, 5 μg/ml insulin, 3 pg/ml triiodothyronine, and 5 μg/ml transferrin. The cells were then incubated at 37 °C with 5% $CO_2$. On the following day, unattached cells were removed, and the growth medium was replaced every 2 days for a period of 6 to 7 days. The prepared renal primary cells exhibited over 70% positivity for proximal tubular cell marker LRP2. Subsequently, the cells were treated with a 100 nM miR-429-3p power inhibitor (Qiagen) for 48 hours following the manufacturer's instructions. This was followed by treatment with 2 μM Cisplatin, with or without 50 μM BT2. After 30 hours of treatment, the levels of miR-429-3p and Acsl4 expression were measured using RT-qPCR.

## Blood urea nitrogen test

The concentrations of plasma urea nitrogen were measured using a diacetyl monoxime methodology-based urea nitrogen direct kit (Stanbio).

## RNA extraction and quantitative PCR

Frozen kidneys were homogenized in liquid nitrogen, solubilized in DNA/RNA shield stabilization solution (Zymo Research), and stored at -80 °C until used. Total RNAs including miRNAs were extracted and then cleaned up with a quick RNA microprep kit (Zymo Research) according to the manufacturer's instructions. Poly (A)-tailed RNAs were reverse transcribed with anchored oligo d(T) primers, d(T)22VN (IDT), using ProtoScript II reverse transcriptase (NEB), while miRNAs were reverse transcribed using the LNA RT kit (Qiagen). SYBR Green-based, real-time quantitative PCR detections with either Luna Universal qPCR Master Mix kit (NEB) for mRNA or LNA miRNA assay kit

(Qiagen) were performed using QuantStudio 6 Pro (Thermo). Quantification and amplification were analyzed with Quantastudio software (Thermo). Gapdh and U6 were used as reference genes for normalization of mRNA and miRNA expression, respectively. The primers and the experimentally validated miRCURY locked nucleic acid (LNA) miRNA PCR assay used in this study are listed in Supplementary Table 1.

## Western analysis

Kidneys were lysed in 50 mM Tris-HCl (pH 7.4)/150 mM NaCl/1% NP-40/ 0.5% sodium deoxycholate/0.1% SDS supplemented with protease inhibitors (Thermo, A32955) by incubating on ice for 30 min. The lysates were sonicated and then cleared by centrifugation at 15,00 × $g$ for 20 min at 4 °C. Protein concentration was measured by the BCA protein assay (Thermo), and aliquots of the cleared lysates were stored at -80 °C until used if needed. The resulting lysates were separated by SDS-PAGE and electrophoretically transferred to a nitrocellulose membrane. The blots were probed with the indicated primary antibodies (Supplementary Table 1): anti-alpha smooth muscle actin (Cell Signaling Technology #19245), anti-fibronectin (Abcam, ab2413), anti-vimentin (Cell Signaling Technology #3932), anti-collagen 1A1 (Cell Signaling Technology #72026), anti-actin (Sigma-Aldrich, A2228), and anti-Acsl4 (Thermo Fisher Scientific, PA5-27137) and then followed by HRP-conjugated secondary antibodies for chemiluminescence detection. The unsaturated signals on the blots were captured and analyzed with iBright FL1000.

## Immunofluorescence

Frozen kidney tissues were rinsed in DPBS and then soaked in 20% sucrose overnight at 4 °C before embedded in OCT and snap-frozen on dry ice. The tissues were then sectioned at a thickness of 10 μm using Microtome cryostat (HM 505 E, MICROM) and placed on SuperfrostPlus glass slides (Fisher). The sections were then fixed with 4% paraformaldehyde and blocked with 10% normal goat serum (Jackson Laboratory). Primary antibodies (Supplementary Table 1) were applied to the blocked sections and left to incubate overnight at 4 °C. The following day, the sections were incubated with secondary antibodies and lectin for 2 hours at room temperature. Nuclear counterstaining was performed using DAPI for 20 minutes. Finally, the sections were mounted with Vectashield® Plus (Vector laboratories) and imaged using an Axio fluorescence microscope (Zeiss) or a STELLARIS confocal microscope (Leica) under the same exposure and light collection conditions over samples. ImageJ was used to export the images as a tiff file. To quantify 4-HNE expression, images were segmented using a thresholding plugin (ImageJ), and the intensities of the foreground were counted and normalized to area and nucleus numbers.

## RNAscope

RNA in situ hybridization was conducted using the RNAscope™ Plus smRNA-RNA HD Reagents Kit (ACD). Briefly, 10 μm cryosections were fixed in 4% paraformaldehyde/DPBS for 1 hour at room temperature. Hybridization of two probes targeting mmu-miR-429-3p and Lrp2 was performed in an RNAscope hybridization oven using the following programs: AMP1 (30 minutes), AMP2 (30 minutes), AMP3 (15 minutes), HRP (15 minutes), S1 Fluorophore (30 minutes), HRP (15 minutes), and C2 Fluorophore (30 minutes). After hybridization, the sections were counterstained with DAPI for visualization.

## Chimeric-eCLIP

Chimeric-miRNA-enhanced cross-linking immunoprecipitation (Chimeric-eCLIP) was described by Van Nostrand et al. and was performed with the assistance of Eclipse Bioinnovations. Briefly, snap-frozen kidneys in liquid nitrogen were pulverized using a mortar and pestle, UV-crosslinked, and lysed to immunoprecipitate miRNA-mRNA-

argonaute2 (Ago2) complexes with anti-Ago2. The bound RNAs in the Ago2 immunoprecipitants were further trimmed with RNA exonuclease (RNase A) and then treated with T4 polynucleotide kinase (3′ phosphatase minus) to encourage ligation between miRNA and mRNA, and then T4 RNA ligase 1 was applied to the donor of ligase RNA with 5′phosphate and the 3′OH end of the RNA acceptor in the immunoprecipitants. The resultant chimeric reads were processed for high-throughput RNA sequencing.

### Total RNA-sequencing, small RNA-sequencing, and their differential expression analysis
RNA sequencing was performed as described[46]. After assessing their quality with BioAnalyzer (Agilent) and Quanti-iT PicoGreen dsDNA assay, RNA libraries were sequenced as 30 million reads, 150-bp paired-end runs on an Illumina HiSeq platform. The adapter sequences in the raw data were trimmed using Cutadapt, and STAR was utilized for aligning paired-end (for total RNA) or single-end (for small RNA) sequences to the mouse genome (mm39) and quantification[47,48]. The differential expression analysis was performed with two-sided DESeq2 (ver 4.2.3)[49].

### chimeric-eCLIP-Seq data analysis
Unique molecular identifiers (umi) were extracted from all read files with umi-tools (ver 11.2) and the adapter was cut using cutadapt (ver 4.1)[47,50]. The sequence files were mapped against RepBase to control artifacts from rRNA. Subsequently, non-repetitive reads were mapped to the mouse reference genome (mm39) using STAR (ver 1.7.10a)[48]. PCR duplicates were removed for aligned reads (non-chimeric reads) using umi-tools, and Ago2 binding site peaks were identified using CLIPper[50–52]. The clusters identified in the Ago2-immunoprecipitant are then normalized against a paired input sample. A peak is defined as a cluster with log2 fold enrichment ≥3 and $p$ value ≤ 0.001. For Chimeric miRNA-RNA reads (unaligned non-repetitive reads), each miRNA was assigned to reads using Bowtie (ver 1.3.1) through mapping the miRNAs[53]. The miRNA portion was trimmed. The remaining mRNA portion was then mapped using STAR[53]. Removal of PCR duplicates and thereafter identification of AGO2 binding sites were performed as similar to non-chimeric reads, and the numbers of chimeric miRNA-mRNA read were obtained.

### Cumulative frequency analysis
mRNA expression data from total RNA-seq were divided based on the presence of target mRNA from chimeric-eCLIP-seq, and the cumulative abundance changes of target mRNAs for each individual miRNA were compared with those of nontarget mRNAs that do not appear in chimeric-eCLIP-Seq but are present in total RNA-Seq. Cumulative frequency distributions were generated with R and the statistical significance of the distribution between non-target mRNA expression and target mRNA expression was determined using the Kolmogorov-Smirnov test.

### GO analysis
To perform statistical enrichment of target genes in a given microRNA from chimeric sequences, ShinyGO 0.76.3[54] was used. Fold enrichment and corrected false discovery rate (FDR) were calculated by comparing a set of predicted target genes of mouse miRNAs from miRDB v 5.0 with a set of genes from chimeric sequences from chimeric chimeric-eCLIP-Seq.

For gene ontology enrichment analysis, significant pathways with overrepresented genes from chimeric mRNA-miRNA sequences were detected using the KEGG (Kyoto Encyclopedia of Genes and Genomes) database. To calculate enrichment FDR (false discovery rate), all genes whose expression was detected from total RNA-seq were used as background.

### Metagene of mRNA sites present in Ago2-chimeric chimeric-eCLIP-Seq
The average number of peaks mapped to the genomic region was generated with MetaPlotR[55]. The number of peaks was calculated for each region of a mapped gene and the average number of peaks for a set number of positions was plotted along the region whose length was normalized.

### Alamar Blue assay
An equal number of BUMPT cells per well were seeded in a 96-well plate with the complete growth medium. After overnight incubation, cells were washed with DPBS and further incubated with 2 μM cisplatin with DMSO as control or 100 μM BT2 in a serum-free medium, respectively. After 15 hours of incubation, the final concentration of 0.04 mg/ml Alamar Blue (AB) was added to each well and the absorbances were measured at 570 and 600 nm for 5 hours every hour, using a microplate reader (BioTek). The percentage of reduction of Alamar Blue was calculated by a formula ((molecular extinction coefficient of AB in the oxidized form at 600 nm $\times$ A$_{570}$ - molar extinction coefficient of AB in the oxidized form at 570 nm $\times$ A$_{600}$) ÷ (molar extinction coefficient of AB in the reduced form at 570 nm $\times$ A′$_{600}$ - molar extinction coefficient of AB in the reduced form at 600 nm x A′$_{570}$, where A$_{600}$ absorbance of test wells at 600 nm; A$_{570}$ absorbance of test wells at 570 nm; A′$_{600}$ absorbance of negative control wells at 600 nm; and A′$_{570}$ absorbance of negative control wells at 570 nm[56].

### Statistical analysis
The statistical significance of the data was evaluated using a two-tailed, unpaired Student's t-test, unless otherwise stated. A threshold of $p$ values < 0.05 was considered statistically significant. No sample size calculation was performed, and no data were excluded from the analysis unless there were sample loss (e.g., mouse death) or indications of issues with experimental steps, as indicated by experimental controls. Multiple biological replicates from independent experiments were used, while technical replicates were not considered independent samples.

### Reporting summary
Further information on research design is available in the Nature Portfolio Reporting Summary linked to this article.

## Data availability
Data supporting the findings of this study are presented within the article and supplemental information. All the raw sequencing datasets generated in this study have been deposited in GEO, Gene Expression Omnibus (https://www.ncbi.nlm.nih.gov/geo/query/acc.cgi?acc=GSE242809) under accession code GSE242809, and the deposited data are publicly available. Source data, including sequencing analysis data, are provided with this paper. Source data are provided with this paper.

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

## Acknowledgements

This work was supported by a fund from the National Institutes of Health grant (R01DK120510, KWON). We thank members of the Kwon lab for their comments on the manuscript.

## Author contributions

S.K. planned the project and obtained funding. H.S., B.L., and S.K. designed experiments. H.S. and B.L. carried out RNA sequencing (total, small, eCLIP-seq), pulldowns, and immunoblotting. S.K. and T.L. carried out sequencing data analysis. H.S., B.L., and S.O. carried out RT-qPCR. H.S. carried out immunofluorescence. H.S. and D.H. carried out miRNA inhibitor experiments. H.S., D.H., and B.L. carried out BT2 experiments. T.L., S.O., and S.K. wrote the manuscript with input from H.S., B.L., and D.H.

## Competing interests

The authors declare no competing interests.
