## [Peer Review File · Nature Communications]

MicroRNA-mediated attenuation of branched-chain amino acid catabolism promotes ferroptosis in chronic kidney diseaseREVIEWER COMMENTS

Reviewer #1 (Remarks to the Author):

In this paper the authors examine the role of microRNAs in cisplatin induced chronic kidney disease. The authors employ a recently developed chimeric-eCLIP approach to define miRNA-mRNA pairs within the RISC that are actively regulating gene expression during injury. Importantly, these studies are performed using kidney tissue rather than cell lines. The authors carefully validate their model of cisplatin induced injury. They validate, maybe for the first time, the use of chimeric-eCLIP is feasible in mouse kidney. Although the use of this technique is new to this application, mapping miRNA-mRNA interactions in models of renal injury is not new (PMID: 28925592). The analysis of sequencing data is clear, and the tools used to analyze the data and determine significance are appropriate. Overall, the study is well-designed.

The data generated suggests that the mRNAs targeted during injury encode genes in common pathways, of which metabolic pathways are highly represented (Fig 2). This suggests that miRNAs work at multiple nodes within pathways to regulate gene expression. Or as the authors state, “multiple miRNAs can regulate a shared group of mRNAs”. If this is the case, it would provide significant insight into how miRNAs regulate gene expression. In particular, it would suggest that single targeting events provide a very incomplete picture of how miRNAs regulate expression during pathology. In this case, the manuscript could be strengthened by further validation showing multimodal regulation on mRNAs by induced miRNAs.

The authors go on to study miR-429-3p one of 11 miRNAs which is induced by cisplatin. It is not clear why this single miRNA was chosen for further study. Given that the authors argue that the focused targeting by miRNAs of mRNAs encoding proteins involved in metabolism may be a major biological feature of how miRNAs work, it would be helpful to provide evidence that at least some of the other predicted interactions are biologically significant. Nevertheless, miR-429-3p appears to target mRNAs that encode proteins involved in BCAA metabolism. Down regulation of these genes by miRNAs contributes to pathology by allowing accumulation of BCAA which lead to ferroptosis. The authors use LNA inhibitors to

show involvement by miR-429-3p and validate this approach. They show that there is an inverse relationship between miR-429 expression and predicted target genes. They go on to show that boosting BCAA catabolism using a small molecule inhibitor BT2 provides an effect like LNA knockdown. Interestingly, this appears to be related to effect of genes involved in ferroptosis. However, the authors do not appear to provide data related to how mi-429-3p knockdown of BT2 treatment effects renal injury. This data is critical. It is also worth considering if the changes observed are generalizable to chronic injury and whether treatment can prevent long term injury.

Reviewer #2 (Remarks to the Author):

The work by Sone et al describes several pairs of miRNAs/mRNAs that may function in cisplatin-induced chronic kidney diseases. They identify miR-429-3p as a driver of renal tubular cell ferroptosis by targeting the mitochondria branched-chain amino acid (BCAA) pathway. They conclude that the BCAA pathway is a potential target for CKD treatment and cisplatin-associated nephrotoxicity.

Several miRNAs including miR-429-3p were identified in this study. They deciphered the role of miR-429-3p in promoting ferroptosis and CKD that provide insight into miRNAs-targeted therapy for CKD. The authors provided amounts of omics data trying to illustrate the regulating role of miRNAs in renal tubular cell ferroptosis in CKD. The presented data are not enough to prove it. More in vivo and ex vivo data including phenotype and mechanism should be provided.

Specific comments:

1. In this study, the author claims ferroptosis, however, we did not see any ferroptosis phenotype except for the decreased expression of Acsl4. ACSL4 is known as an upstream negative regulator of ferroptosis, it is also a protein for antioxidation. It is arbitrary to conclude ferroptosis without information on whether cells are in the peroxidation status, especially when another ferroptosis regulator Gpx4 is stable.
2. This work focused on identifying novel miRNAs in regulating cisplatin-induced chronic kidney diseases. They show that miR-429-3p acts a role in renal tubular cell ferroptosis and CKD. In the next confirmative experiment, they just showed that miR-429-3p inhibitor could

reduce *Acsl4* expression. Could the authors provide more gain or loss-of-function data to prove the existence of ferroptosis?

3. The authors established a CKD mouse model by using cisplatin, is there an *ex vivo* model to mimic CKD using renal tubular cell line to prove your conclusion?

4. As we know, cisplatin is a nephrotoxic drug with widespread detrimental effects including disrupting mitochondrial oxidative phosphorylation (OXPHOS). We are not surprised to see that cisplatin impaired mitochondria branched-chain amino acid (BCAA) pathway. The provided data are not enough to support the indispensable role of miR-429-3p in mediating the cisplatin-impaired BCAA pathway.

5. The author should initially confirm the existence and extent of cell ferroptosis in CKD before testing the protective effect of the BCAA inhibitor BT2. We believe cell ferroptosis is remarkable at the early stage of kidney injury in response to cisplatin. Compared to ferroptosis, cell repair, and fibrosis are dominant at the late resolving stage.

6. Each miRNA can target a range of mRNAs simultaneously, the unpredictable and multiple outcomes limit the significance of your findings.

7. Many miRNAs are differentially expressed after cisplatin treatment. the author did not confirm these miRNAs are the driver instead of the marker for ferroptosis.

8. The data suggest that cisplatin-induced miR-429-3p represses mRNA expressions of key components of the BCAA catabolic pathway. The specific mRNAs were not defined, which will undermine the significance of the study.

Above all, the results described in this manuscript are not sufficient to publish in Nature Communications in its current version.

Reviewer #3 (Remarks to the Author):

The manuscript by Sone et al. describes the identification of microRNA targets in chronic kidney disease models, and analysis of these miRNA:target interactions to discover regulatory nodes in this disease. The authors are one of the few studies to utilize large-scale experimental mapping of miRNA interactions in *in vivo* contexts, and the identification of miRNA control of this disease is an interesting avenue for future research.

As a first comment: every reference to chimeric eCLIP in the manuscript seems like it should

reference the Manakov et al.

<https://www.biorxiv.org/content/10.1101/2022.02.13.480296v2> chimeric eCLIP pre-print; the referenced papers 15-16 do not seem to discuss chimeric eCLIP at all. The fact that that initial work also described chimeric eCLIP from tissue (liver) seems like it's also highly relevant to the discussion paragraph that implies chimeric eCLIP from tissues has not been done. More broadly, the manuscript seems to treat chimeric eCLIP as the first chimeric miRNA target identification method; but this ignores the initial chimeric CLIP/CLASH publications, some of which were also initially described from mouse tissues (e.g. Moore et al. Nat Comms 2015 PMID 26602609 performed chimeric CLIP from mouse brain, and Grosswendt et al. Mol Cell 2014 PMID 24857550 performed chimeric re-analysis of AGO2 HITS-CLIP from mouse brain). I think the novelty claims here need to be toned down a bit, and the intro/discussion need to include more of the prior literature in this area.

There also is not enough specificity throughout, in description of the data, experiments performed, and supplemental tables – For example, Sup Table 4, I can't find any description of what the numbers actually represent. Figure 2b, I can't find any description of what the criteria and cutoffs are here – do they have to be seen in multiple replicates? Is there a minimum read number cutoff? Sup. Fig. 2d, the callout in the text says “cisplatin-induced enhancement of Ago2 association was individually confirmed by RT-qPCR for select miRNAs” which implies that this was done on AGO2-immunoprecipitated samples, but the figure legend seems to suggest it's just RT-qPCR of tissue samples, which is it?

Additionally, I think many places are worded too strongly given the analyses shown - like the text callout to Fig 2b “further identified that Ago2 binds to 19.68% of mRNA and 40.90% of miRNA that are expressed in the kidney” - does AGO2 actually only bind to 20% of mRNAs, or is this simply the number of binding sites seen at the limited sequencing depth performed? What does ‘differentially associated with AGO2’ mean (it seems like the authors are arguing that nearly half of genes with chimeric reads are differentially enriched, which to me seems more likely to reflect a high false discovery rate than a true result; but with no FDR-type analysis in the manuscript it's hard to tell). Similarly, “we noted that 11 of the 19 miRNAs significantly reduced the expressions of their target mRNAs compared with those of nontarget mRNAs” should be written far more cautiously, e.g. “we noted that for 11 of the

19 miRNAs their targets showed significantly reduced expression upon cisplatin treatment compared with those of nontarget mRNAs” – other than the miR-429-3p experiment in Fig. 4, the paper doesn’t prove (or even really test) that those 11 miRNAs directly reduce expression of those targets.

To me, Figure 5 seems disconnected from the paper – the lead-in says that ‘BAC takes a significant share of the genes’, but the experiments that follow then only test the effect of BT2 – it seems to be like for this figure to be directly connected to the rest of the paper, one would want to see some sort of suppression / epistatic interaction between BT2 and miR-429-3p inhibitor. (I also don’t see where that ‘significance’ claim is substantiated beyond a labeling of 5-8 genes in the bottom of Fig 3c – seems like one would want some test here to show that the BAC genes are enriched relative to expectation)

Other comments:

I’m a bit confused by the idea of Figure 2e – the idea of looking at co-targeting by different miRNAs is interesting, but why not perform the analysis (or at least validate it) using chimeric eCLIP targets for all the miRNAs instead of using chimeric targets for one miR and then looking at miRDB predicted targets for the putative co-targeting miRs?

The ecdf plots (Fig. 3a and 4d) really should have x-axis go from only -1 to +1 or -1.5 to +1.5; as plotted it’s very hard to see any differences

REVIEWER COMMENTS

Reviewer #1 (Remarks to the Author):

In this paper the authors examine the role of microRNAs in cisplatin induced chronic kidney disease. The authors employ a recently developed chimeric-eCLIP approach to define miRNA-mRNA pairs within the RISC that are actively regulating gene expression during injury. Importantly, these studies are performed using kidney tissue rather than cell lines. The authors carefully validate their model of cisplatin induced injury. They validate, maybe for the first time, the use of chimeric-eCLIP is feasible in mouse kidney. Although the use of this technique is new to this application, mapping miRNA-mRNA interactions in models of renal injury is not new (PMID: 28925592). The analysis of sequencing data is clear, and the tools used to analyze the data and determine significance are appropriate. Overall, the study is well-designed.

We appreciate the reviewer's kind comments on our manuscript. While our attempt to map miRNA-mRNA interactions in kidney injury may not be unprecedented, we would like to emphasize that the study the reviewer mentioned (PMID: 28925592) used a qualitatively different approach from ours.

First, the study (PMID: 28925592) employed "PAR-CLIP" to profile miRNAs and mRNAs that are associated with Ago2. PAR-CLIP is a variation of the CLIP-seq methodology to identify RNA molecules associated with an RNA-binding protein such as Ago2. In particular, the PAR-CLIP approach that pulls down Ago2-associated RNAs cannot determine pairs of "directly interacting" miRNA and mRNA. However, chimeric eCLIP-seq was deliberately developed to uncover direct pairs of miRNA and its cognate target mRNAs, both of which are together complexed with Ago2.

Second, the study (PMID: 28925592) utilized the human kidney proximal tubule-originated cell line HK-2 to identify Ago2-associated miRNAs and mRNAs following cell injury. In contrast, our current study is distinct in that we analyzed the mouse kidney itself, rather than cultured cells, to investigate miRNA-mediated gene expression during chronic kidney injury.

The data generated suggests that the mRNAs targeted during injury encode genes in common pathways, of which metabolic pathways are highly represented (Fig 2). This suggests that miRNAs work at multiple nodes within pathways to regulate gene expression. Or as the authors state, "multiple miRNAs can regulate a shared group of mRNAs". If this is the case, it would provide significant insight into how miRNAs regulate gene expression. In particular, it would suggest that single targeting events provide a very incomplete picture of how miRNAs regulate expression during pathology. In this case, the manuscript could be strengthened by further validation showing multimodal regulation on mRNAs by induced miRNAs.

We appreciate the thoughtful evaluation of our findings. As the reviewer pointed out, a single miRNA can simultaneously target numerous mRNAs, and likewise, a given mRNA or pathway

can be regulated by several miRNAs. These are well documented and conceptualized in the miRNA field, and our data revealed and confirmed the same phenomenon (e.g., Figs. 3b-3c).

In this study, we were interested in identifying a novel aspect of miRNA-mediated gene expression alteration in chronic kidney injury. We therefore focused on BCAA catabolism, as our chimeric eCLIP-seq unveiled it as a pathway uniquely targeted by certain cisplatin-induced miRNAs, such as miR-429-3p and miR-21a-5p. We could have validated all cisplatin-induced miRNAs and all possible pathways under miRNA regulation. However, we are afraid that such efforts seem impractical and beyond the scope of the current study.

The authors go on to study miR-429-3p one of 11 miRNAs which is induced by cisplatin. It is not clear why this single miRNA was chosen for further study. Given that the authors argue that the focused targeting by miRNAs of mRNAs encoding proteins involved in metabolism may be a major biological feature of how miRNAs work, it would be helpful to provide evidence that at least some of the other predicted interactions are biologically significant. Nevertheless, miR-429-3p appears to target mRNAs that encode proteins involved in BCAA metabolism. Down regulation of these genes by miRNAs contributes to pathology by allowing accumulation of BCAA which leads to ferroptosis. The authors use LNA inhibitors to show involvement by miR-429-3p and validate this approach. They show that there is an inverse relationship between miR-429 expression and predicted target genes. They go on to show that boosting BCAA catabolism using a small molecule inhibitor BT2 provides an effect like LNA knockdown. Interestingly, this appears to be related to the effect of genes involved in ferroptosis. However, the authors do not appear to provide data related to how miR-429-3p knockdown or BT2 treatment affects renal injury. This data is critical. It is also worth considering if the changes observed are generalizable to chronic injury and whether treatment can prevent long term injury.

We thank the reviewer for highlighting the main findings of our study. As indicated by the reviewer, miR-429-3p constitutes 11 miRNAs inducible by cisplatin in the kidney. We chose to investigate this microRNA, as our chimeric eCLIP-seq revealed that miR-429-3p, along with miR-21a-5p, appears to target mRNAs involved in BCAA catabolism, while the other 9 cisplatin-induced miRNAs did not show clear association with BCAA catabolism. We could have studied both miR-429-3p and miR-21a-5p, but, as stated above, studying both the miRNAs would be impractical, considering huge cost and efforts (e.g., LNA studies, RNA-seq, mouse experiments, etc. all cost substantially). We thought that focusing on miR-429-3p, as a first case, would enable us to establish that ferroptosis is regulated by miRNA via modulating the expression of the genes involved in BCAA catabolism.

We would like to inform the reviewer that a recent study reported that miR-21a-5p can stimulate fibrosis by altering metabolism in the injured kidney and that we cited this study in the Discussion section (Ref. 41). It should be noted that this study (Ref. 41) did not show whether miR-21a-5p is involved in BCAA catabolism (as well as ferroptosis); but simply revealed its association with metabolism.

Although we did not study directly on miR-21a-5p, our revised manuscript now includes a new result that supports an additional factor other than miR-429-3p, such as miR-21a-5p, can regulate ferroptosis via modulating BCAA catabolism (please note the new Supplementary Fig. 5). By using an ex vivo culture of cisplatin-injured mouse kidney cells, we now demonstrate that the decrease in Ascl4 expression by the LNA inhibiting miR-429-3p can be further reduced by addition of BT2 (Supplementary Fig. 5d), suggesting that cellular BCAA level can be modulated not only by miR-429-3p but also by another source such as miR-21a-5p.

In addition, our revised manuscript presents a new observation that cisplatin can trigger an increase of lipid peroxidation in the kidney, which is indispensable for the execution of ferroptosis (new Fig. 4f). We demonstrate that inhibition of miR-429-3p by LNA (new Fig. 4g) and administration of BT2 (new Fig. 5b) can reverse the increase of lipid peroxidation in response to cisplatin, as evidenced by immunofluorescence microscopy of the lipid peroxidation marker 4-HNE.

Furthermore, the combined effect of BT2 and the LNA miR-429-3p inhibitor on the decrease of Ascl4 level as mentioned above (new Supplementary Fig. 5d) appears to be associated with correlative reduction of the lipid peroxidation marker 4-HNE (new Supplementary Fig. 5e), again suggesting the presence of another contributor to the level of BCAA, which would help engage cisplatin-inducible ferroptosis.

We appreciate the reviewer's concerns, but assessing the long-term, injury-preventing effect of BT2 treatment in multiple kidney injury models seems beyond the scope of the current study. However, we find this point intriguing and plan to pursue it in future research. Thank you for the suggestion.

Reviewer #2 (Remarks to the Author):

The work by Sone et al describes several pairs of miRNAs/mRNAs that may function in cisplatin-induced chronic kidney diseases. They identify miR-429-3p as a driver of renal tubular cell ferroptosis by targeting the mitochondria branched-chain amino acid (BCAA) pathway. They conclude that the BCAA pathway is a potential target for CKD treatment and cisplatin-associated nephrotoxicity.

Several miRNAs including miR-429-3p were identified in this study. They deciphered the role of miR-429-3p in promoting ferroptosis and CKD that provide insight into miRNAs-targeted therapy for CKD. The authors provided amounts of omics data trying to illustrate the regulating role of miRNAs in renal tubular cell ferroptosis in CKD. The presented data are not enough to prove it. More in vivo and ex vivo data including phenotype and mechanism should be provided.

We thank the reviewer for the comments on our work. Strengthening our conclusions, we now present a revised manuscript that includes several new lines of data. These include:

- Direct measurement of lipid peroxidation with 4-HNE immunofluorescence (new Figs 4f-4g, Fig. 5b, Supplementary Fig. 5e)
- An ex vivo culture of kidney cells to compare the range of ferroptosis marker expression between miR-429-3p inhibitor alone and in combination with BT2 (new Supplementary Fig. 5)
- RNAscope™ in situ hybridization to measure cisplatin-inducibility of miR-429-3p expression in the kidney (new Supplementary Fig. S2e).

Specific comments:

1. In this study, the author claims ferroptosis, however, we did not see any ferroptosis phenotype except for the decreased expression of *Acs14*. *ACSL4* is known as an upstream negative regulator of ferroptosis, it is also a protein for antioxidation. It is arbitrary to conclude ferroptosis without information on whether cells are in the peroxidation status, especially when another ferroptosis regulator *Gpx4* is stable.

We thank the reviewer for the comment. We first would like to remind the reviewer that *Acs14* “promotes” ferroptosis (“not inhibits” ferroptosis), as it alters the cellular lipid composition to increase the level of PUFA (polyunsaturated fatty acids) (PMID: 27842070). We also would like to point out that *GPX4* is a negative component of the ferroptosis pathway (PMID: 24439385), and that a growing number of studies are recently reporting ferroptosis regulation *irrespective* of *GPX4* (PMID: 31634899, 37267948, 33981038), which is consistent with our findings.

Our revised manuscript now presents new data that support our observations of reduced *Acs14* expression elicited by the LNA miR-429-3p inhibitor (Fig. 4e) and BT2 (Fig. 5a) in the cisplatin-injured kidney. We measured the extent of lipid peroxidation, the increase of which is indicative of ferroptosis, by using immunofluorescence microscopy for 4-HNE, a lipid peroxidation marker. We demonstrate that administration of either the LNA miR-429-3p inhibitor (new Fig. 4g), BT2

(new Fig. 5b), or both (new Supplementary Fig. 5e) can decrease the level of 4-HNE, which otherwise remains elevated in response to cisplatin.

2. This work focused on identifying novel miRNAs in regulating cisplatin-induced chronic kidney diseases. They show that miR-429-3p acts a role in renal tubular cell ferroptosis and CKD. In the next confirmative experiment, they just showed that miR-429-3p inhibitor could reduce Acsl4 expression. Could the authors provide more gain or loss-of-function data to prove the existence of ferroptosis?

Thank you for your comment. As mentioned in Specific Comment #1, our revised manuscript now presents that lipid peroxidation, as a direct indicator of ferroptosis using 4-HNE immunostaining, is induced by cisplatin and can be reduced through the inhibition of miR-429-3p or the addition of BT2. Please refer to the revised text and new figures (new Figs, 4f-4g and 5b, and Supplementary Fig. 5e). We conducted this analysis in both the kidney organ and ex vivo kidney-derived cultured cells. Additionally, the Alamar Blue Assay presented in Fig. 5d, using the kidney-originated immortalized cell line BUMPT, provides further evidence that BT2 can reverse ferroptotic cell death.

3. The authors established a CKD mouse model by using cisplatin, is there an ex vivo model to mimic CKD using renal tubular cell line to prove your conclusion?

Thank you for the comment. As the reviewer suggested, we performed an ex vivo experiment, in which cells (mostly with LRP2-positive kidney proximal tubule characteristics; new Supplementary Fig. 5b) were derived from the mouse kidney and treated with cisplatin. We observed that the LNA miR-429-3p inhibitor can reduce the expression of Acsl4, which can be further reduced by BT2 in this ex vivo cultured cell model (new Supplementary Fig. 5d). Likewise, the accumulation of the lipid peroxidation marker 4-HNE can be reduced correspondingly (new Supplementary Fig. 5e).

4. As we know, cisplatin is a nephrotoxic drug with widespread detrimental effects including disrupting mitochondrial oxidative phosphorylation (OXPHOS). We are not surprised to see that cisplatin impaired mitochondria branched-chain amino acid (BCAA) pathway. The provided data are not enough to support the indispensable role of miR-429-3p in mediating the cisplatin-impaired BCAA pathway.

We agree that the impaired oxidative phosphorylation of mitochondria caused by cisplatin results in numerous deleterious cellular changes, including perturbation of BCAA catabolism. However, as far as we know, our current study is an unprecedented experimental demonstration, establishing the first case that links BCAA catabolic mitigation to ferroptosis engagement via gene expression alteration, which is mediated by select, cisplatin-inducible miRNAs such as miR-429-3p.

Furthermore, we would like to emphasize that we do not argue that miR-429-3p plays an indispensable role in mounting ferroptosis in the cisplatin-injured kidney; we argue instead that miR-429-3p can partially contribute to the ferroptotic response. As stated in the text, our data indicate that “miR-21a-5p” is another cisplatin-induced miRNA that appears to target BCAA catabolism (Fig. 3c). Supporting this possible ferroptosis-inducing route alternative to that by miR-429-3p, the new Supplementary Fig. 5 demonstrates that the LNA miR-429-3p inhibitor-mediated decrease of *Acsl4* expression and 4-HNE appearance can be further reduced by the addition of BT2, suggesting that miR-429-3p can control only part of the cisplatin-induced BCAA catabolism and ensuing ferroptosis.

5. The author should initially confirm the existence and extent of cell ferroptosis in CKD before testing the protective effect of the BCAA inhibitor BT2. We believe cell ferroptosis is remarkable at the early stage of kidney injury in response to cisplatin. Compared to ferroptosis, cell repair, and fibrosis are dominant at the late resolving stage.

Our new data demonstrate the existence of ferroptosis in the cisplatin-injured kidney in the absence of BT2. Specifically, measuring the lipid peroxidation status with 4-HNE reveals a marked up-regulation of lipid peroxidation, indicative of ferroptosis, in the cisplatin-injured kidney, which remains observable at the end of the 7-week period (new Fig. 4f). The addition of either the LNA miR-429-3p inhibitor (new Fig. 4g), BT2 (new Fig. 5b), or both (new Supplementary Fig. 5e) reverses this induction of ferroptosis.

We agree with the reviewer that ferroptosis may occur in the early stages of CKD. However, we would like to emphasize that ferroptosis persists at the end of the 7-week experimental period in our study. In addition, when compared to two prevalent fibrosis models, UUO and unilateral IRI, the cisplatin model we used develops much less prominent fibrosis. It would be interesting to investigate how suppressed ferroptosis impacts fibrosis in this model. Future studies can explore how cells mount various responses in succession by employing kinetic and longitudinal analyses, which would be of great interest.

6. Each miRNA can target a range of mRNAs simultaneously, the unpredictable and multiple outcomes limit the significance of your findings.

It is widely accepted and has already been conceptualized that each miRNA can target numerous mRNA targets and that a given mRNA or a pathway can be regulated by several miRNAs. Our data revealed and confirmed the same phenomenon (e.g., Figs. 3b-3c). Therefore, the significance of our findings cannot be limited, considering this consensus of the miRNA field.

In this study, our goal was to uncover a novel aspect of miRNA-mediated gene expression alteration in chronic kidney injury. We specifically focused on BCAA catabolism because our chimeric eCLIP-seq analysis revealed it as a pathway uniquely targeted by certain cisplatin-

induced miRNAs, such as miR-429-3p and miR-21a-5p. While it is possible to validate all cisplatin-induced miRNAs and explore all potential pathways under miRNA regulation, such an endeavor would be impractical, due to the substantial costs and efforts involved in LNA experiments, mouse experiments, and more, and thus beyond the scope of our current study.

7. Many miRNAs are differentially expressed after cisplatin treatment. the author did not confirm these miRNAs are the driver instead of the marker for ferroptosis.

Our investigation did not aim to identify a miRNA that 'drives' ferroptosis in cisplatin-injured kidneys. Our data show that miRNA 429-3p targets the mRNAs involved in 'BCAA catabolism', but it does not directly target genes associated with 'ferroptosis,' such as *Acsl4* and *Tfrc*. The question of how ferroptosis pathway genes, like *Acsl4* and *Tfrc*, can be regulated through the repression of BCAA catabolism genes remains a subject for future study.

8. The data suggest that cisplatin-induced miR-429-3p represses mRNA expressions of key components of the BCAA catabolic pathway. The specific mRNAs were not defined, which will undermine the significance of the study.

Above all, the results described in this manuscript are not sufficient to publish in Nature Communications in its current version.

We thank the reviewer for pointing out this. We apologize for not distinctly presenting the list of the miR-429-3p-regulated mRNAs in our initial submitted manuscript. In our revised manuscript, we now present the miR-429-3p-regulated mRNA list in a separate, dedicated tab. Please find the revised “Supplementary Table 5: Total RNA-Seq: LNA miR-429-3p inhibitor administration (mRNAs) and target mRNA list of miR-429-3p.” For your convenience, we have also copied the BCAA catabolic pathway targeted by miR-429-3p below.

Type	Gene	Log2 FC	Adj.p-value
miR-429-3p target identified by eCLIP that are associated with BCAA	Aldh6a1	-0.84	< 0.001
	Oxct1	-2.13	< 0.001
	Ehhadh	-0.86	< 0.001
	Hadh	-0.97	< 0.001
	Hibadh	-1.13	< 0.001
	Abat	-0.79	< 0.001

Reviewer #3 (Remarks to the Author):

The manuscript by Sone et al. describes the identification of microRNA targets in chronic kidney disease models, and analysis of these miRNA:target interactions to discover regulatory nodes in this disease. The authors are one of the few studies to utilize large-scale experimental mapping of miRNA interactions in in vivo contexts, and the identification of miRNA control of this disease is an interesting avenue for future research.

As a first comment: every reference to chimeric eCLIP in the manuscript seems like it should reference the Manakov et al. <https://www.biorxiv.org/content/10.1101/2022.02.13.480296v2> chimeric eCLIP pre-print; the referenced papers 15-16 do not seem to discuss chimeric eCLIP at all. The fact that that initial work also described chimeric eCLIP from tissue (liver) seems like it's also highly relevant to the discussion paragraph that implies chimeric eCLIP from tissues has not been done. More broadly, the manuscript seems to treat chimeric eCLIP as the first chimeric miRNA target identification method; but this ignores the initial chimeric CLIP/CLASH publications, some of which were also initially described from mouse tissues (e.g. Moore et al. Nat Comms 2015 PMID 26602609 performed chimeric CLIP from mouse brain, and Grosswendt et al. Mol Cell 2014 PMID 24857550 performed chimeric re-analysis of AGO2 HITS-CLIP from mouse brain). I think the novelty claims here need to be toned down a bit, and the intro/discussion need to include more of the prior literature in this area.

We thank the reviewer for the kind evaluations of our work. We appreciate the comments on various CLIP-seq studies that should be cited in our manuscript. Now the Manakov biorxiv study is referenced in our revised manuscript, which also cites all the references more appropriate to the methodology that the reviewer kindly reminded us with. As the reviewer suggested, we also toned down our words to respect those preceding chimeric eCLIP studies.

There also is not enough specificity throughout, in description of the data, experiments performed, and supplemental tables – For example, Sup Table 4, I can't find any description of what the numbers actually represent. Figure 2b, I can't find any description of what the criteria and cutoffs are here – do they have to be seen in multiple replicates? Is there a minimum read number cutoff? Sup. Fig. 2d, the callout in the text says “cisplatin-induced enhancement of Ago2 association was individually confirmed by RT-qPCR for select miRNAs” which implies that this was done on AGO2-immunoprecipitated samples, but the figure legend seems to suggest it's just RT-qPCR of tissue samples, which is it?

Thank you for the comments. We apologize for the unclarity. Please find the following explanations:

Sup Table 4, I can't find any description of what the numbers actually represent.

The numbers indicate 'chimeric peaks adjusted by the length of peaks' This description is added in the Supp Table 4.

Figure 2b, I can't find any description of what the criteria and cutoffs are here – do they have to be seen in multiple replicates? Is there a minimum read number cutoff?

To minimize the false positive signal in identifying enriched peaks, we employed a p-value threshold of <0.001, along with a fold change threshold of 8 (described in the Method section). When we examined the FDR-adjusted p-values, all of the identified peaks remained significant, even with an adjusted threshold of 0.01. We chose a conservative cutoff, but it is possible that additional true peaks exist.

Sup. Fig. 2d, the callout in the text says “cisplatin-induced enhancement of Ago2 association was individually confirmed by RT-qPCR for select miRNAs” which implies that this was done on AGO2-immunoprecipitated samples, but the figure legend seems to suggest it's just RT-qPCR of tissue samples, which is it?

The data is from RT-qPCR with total RNAs, and we apologize for this mistake. It has been corrected in the revised manuscript.

Additionally, I think many places are worded too strongly given the analyses shown - like the text callout to Fig 2b “further identified that Ago2 binds to 19.68% of mRNA and 40.90% of miRNA that are expressed in the kidney” - does AGO2 actually only bind to 20% of mRNAs, or is this simply the number of binding sites seen at the limited sequencing depth performed? What does ‘differentially associated with AGO2’ mean (it seems like the authors are arguing that nearly half of genes with chimeric reads are differentially enriched, which to me seems more likely to reflect a high false discovery rate than a true result; but with no FDR-type analysis in the manuscript it's hard to tell). Similarly, “we noted that 11 of the 19 miRNAs significantly reduced the expressions of their target mRNAs compared with those of nontarget mRNAs” should be written far more cautiously, e.g. “we noted that for 11 of the 19 miRNAs their targets showed significantly reduced expression upon cisplatin treatment compared with those of nontarget mRNAs” – other than the miR-429-3p experiment in Fig. 4, the paper doesn't prove (or even really test) that those 11 miRNAs directly reduce expression of those targets.

Thank you for the comments. Please note that as you may know, chimeric eClip-seq is inherently inefficient. Considering the difficulty of achieving a substantial degree of UV crosslinking between Ago2 and associated mRNA and miRNA and of intermolecular ligation between mRNA and miRNA, among many other factors. Please find the following explanations:

does AGO2 actually only bind to 20% of mRNAs, or is this simply the number of binding sites seen at the limited sequencing depth performed? What does ‘differentially associated with AGO2’ mean (it seems like the authors are arguing that nearly half of genes with chimeric reads are differentially enriched, which to me seems more likely to reflect a high false discovery rate than a true result; but with no FDR-type analysis in the manuscript it's hard to tell).

We think it is due to limited sequencing depth. Accordingly, we revised the sentences.

We apologize for the mistake in the figure, as it was incorrect. Thank you for bringing this mistake to our attention. The percentages in the figure originally represented the ratio of the indicated RNAs detected in chimeric-eCLIP-seq to those in total RNA-seq or small RNA-seq.

However, due to differences in the total numbers of differentially expressed (DE) and non-DE RNAs between total RNA-seq and small RNA-seq, there is no one-to-one correspondence between miRNA and RNA interactions, and the comparison was inaccurate. Because of the erroneous figure and resulting confusion in the description, we have decided to remove the relevant sentences. Nevertheless, to address the coverage of our chimeric-eCLIP-seq compared to bulk RNA-seq, we updated the figure.

The P-values presented in the supplementary data were calculated using the DESeq2 package and represent FDR-adjusted p-values. In the context of the differential expression analyses, we applied the following filtering criteria: a minimum mRNA count threshold of 10 and a miRNA count threshold of 5, with the stipulation that at least two samples out of the total of three samples per group met these criteria."

Similarly, "we noted that 11 of the 19 miRNAs significantly reduced the expressions of their target mRNAs compared with those of nontarget mRNAs" should be written far more cautiously, e.g. "we noted that for 11 of the 19 miRNAs their targets showed significantly reduced expression upon cisplatin treatment compared with those of nontarget mRNAs" – other than the miR-429-3p experiment in Fig. 4, the paper doesn't prove (or even really test) that those 11 miRNAs directly reduce expression of those targets.

We agree and have updated the revised manuscript accordingly. Thank you for the correction.

While validating all cisplatin-induced miRNAs and their potential pathways is theoretically possible, it becomes impractical due to the substantial costs and efforts involved in LNA experiments, mouse experiments, and more. Furthermore, our study primarily centers on the regulation of BCAA catabolism in relation to ferroptosis in cisplatin-injured kidneys, making a comprehensive investigation of all cisplatin-induced miRNAs beyond the scope of our current research.

To me, Figure 5 seems disconnected from the paper – the lead-in says that 'BAC takes a significant share of the genes', but the experiments that follow then only test the effect of BT2 – it seems to be like for this figure to be directly connected to the rest of the paper, one would want to see some sort of suppression / epistatic interaction between BT2 and miR-429-3p inhibitor. (I also don't see where that 'significance' claim is substantiated beyond a labeling of 5-8 genes in the bottom of Fig 3c – seems like one would want some test here to show that the BAC genes are enriched relative to expectation)

We thank the reviewer for the constructive comment. The revised manuscript now presents a sort of experiment that the reviewer suggested. Please find the text and the new Supplementary Fig. 5. We performed an ex vivo experiment, in which cells (mainly with LRP2-positive kidney proximal tubule characteristics; new Supplementary Fig. 5b) were derived from the mouse kidney and treated with the LNA miR-429 3p inhibitor and cisplatin in the presence or absence of BT2. We observed that the LNA miR-429-3p inhibitor can reduce the expression of *Acs14*,

which can be further reduced by BT2 in this ex vivo cultured cell model (new Supplementary Fig. 5d). Similarly, the appearance of the 4-HNE lipid peroxidation marker, which is indicative of ferroptosis, can be reduced accordingly (new Supplementary Fig. 5e).

When comparing the cisplatin-induced mRNA profiles obtained from (1) the total RNA-seq and (2) the chimeric eCLIP-seq (target mRNA) in 'Fig. 3b', it becomes evident that GO terms such as 'valine, leucine, and isoleucine degradation' are disproportionately represented in the target mRNAs, not in the total mRNAs. Note that valine, leucine, and isoleucine are collectively referred to as branched-chain amino acids (BCAA). This figure demonstrates significant enrichment of BAC (BCAA catabolism) genes in the chimeric eCLIP-seq datasets.

Subsequently, we conducted a search for BAC genes targeted by individual miRNAs, which led to the identification of miR-429-3p and miR-21-5p, as described in the text. Please also find our updated Supplementary Table 5. We now added the miR-429-3p-regulated mRNA list in a separate, dedicated tab.

Other comments:

I'm a bit confused by the idea of Figure 2e – the idea of looking at co-targeting by different miRNAs is interesting, but why not perform the analysis (or at least validate it) using chimeric eCLIP targets for all the miRNAs instead of using chimeric targets for one miR and then looking at miRDB predicted targets for the putative co-targeting miRs?

We apologize for not clearly explaining our analysis in Fig. 2e in the text. In this analysis, we compared a given miRNA chimeric-eCLIP-seq dataset with existing miRNA target sets from miRDB. The purpose of this analysis is twofold. First, we aimed to determine whether the ligations between miRNA and mRNA associated with pulled Ago2 proteins are inter-Ago or intra-Ago, a critical step in identifying direct miRNA:mRNA interactions. Second, we assessed how well the experimental chimeric dataset aligns with the corresponding predicted miRDB dataset. The results indicate that the chimeric eCLIP dataset is highly likely from intra-Ago ligation and well-represented in the corresponding miRDB set. We could have compared target compositions between chimeric-eCLIP-seq datasets, but we wanted to employ datasets that are already widely used as reference to compare mRNA targets of a miRNA such as miRDB.

The ecdf plots (Fig. 3a and 4d) really should have x-axis go from only -1 to +1 or -1.5 to +1.5; as plotted it's very hard to see any differences

We thank the reviewer for the suggestion. We have updated the corresponding figures to make them more intelligible to the readers. Please find the new Figs. 3a and 4d.

REVIEWER COMMENTS

Reviewer #2 (Remarks to the Author):

The revised manuscript has been significantly improved. I have no more concerns.

Reviewer #3 (Remarks to the Author):

I appreciate the author's responses and edits to my previous comments

I appreciate the additional experiment in Sup. Fig. 5d-e, though unfortunately I don't think it adds much – the lack of a BT2-only sample makes it impossible for me to tell whether the miRNA-429-3p inhibitor provides an additive effect or what's observed is simply the effect of BT2 alone; either way, I'm not really sure how an additive effect would prove that these are acting through the same pathway (I'd more expect the opposite – if BT2 alone gives the same effect as BT2 + 429-3p inhibitor, I think that's a stronger argument that BT2 is acting as a 429-3p inhibitor in this context, but that condition is missing here)

Other comments:

The size of the bars in Fig 2b don't seem to have any correspondence to the number values

Pg 9 “were largely completed within the miR-143-3p-containing miRISC, rather than between various miRISCs” – I think this would be more simply stated as ‘were enriched for real interactions rather than background artifacts’, because I don't think that the miRDB target set analysis really tests anything about them being ‘between miRISCs’ (versus say simply free mRNA).

Refs on pg 8 “we employed chimeric-eCLIP-seq^{27,28}” should just be ref #16; I'm not really sure how either of 27 or 28 are relevant here

I'd still like a bit more clarity in Sup Table 4 – I still don't totally understand what those numbers actually are (the most straightforward interpretation is that it's the # of chimeric

peaks identified at that gene divided by the length in nt? But that seems like a very strange metric – it seems like just the # of chimeric peaks would be more useful to provide)

REVIEWER COMMENTS

Reviewer #2 (Remarks to the Author):

The revised manuscript has been significantly improved. I have no more concerns.

Reviewer #3 (Remarks to the Author):

I appreciate the author's responses and edits to my previous comments

I appreciate the additional experiment in Sup. Fig. 5d-e, though unfortunately I don't think it adds much – the lack of a BT2-only sample makes it impossible for me to tell whether the miRNA-429-3p inhibitor provides an additive effect or what's observed is simply the effect of BT2 alone; either way, I'm not really sure how an additive effect would prove that these are acting through the same pathway (I'd more expect the opposite – if BT2 alone gives the same effect as BT2 + 429-3p inhibitor, I think that's a stronger argument that BT2 is acting as a 429-3p inhibitor in this context, but that condition is missing here)

We thank the reviewer for the comment. We would like to mention that Supplementary Figure 5 was inquired by another reviewer. In the revised manuscript, we have now included the control of BT2 only as you requested. We note that combined treatment of cells with BT2 and the miR-429-3p LNA inhibitor resulted in an enhanced reduction of both Ascl4 expression (Supplementary Fig. 5d) and peroxidized lipid 4-HNE levels (Supplementary Fig. 5e) in response to cisplatin, surpassing the effects of individual treatments with either BT2 or the LNA inhibitor. Our interpretation of the results is shown in the revised manuscript (highlighted on page 17) as follows.

“These results suggest that additional miRNAs (e.g., miR-21a-5p) can inhibit BCAA catabolism together with miR-429-3p to promote ferroptosis. The additive effect of the combined treatment of BT2 and the miR-429-3p LNA inhibitor further suggests that additional mechanisms, which depend exclusively either on BCAA catabolism or on miR-429-3p but do not operate in the miR-429-3p-induced BCAA catabolic attenuation pathway, can also contribute to cisplatin-induced ferroptosis.”

Other comments:

The size of the bars in Fig 2b don't seem to have any correspondence to the number values

We thank the reviewer for pointing out this. We updated Figure 2 as you suggested.

Pg 9 “were largely completed within the miR-143-3p-containing miRISC, rather than between various miRISCs” – I think this would be more simply stated as ‘were enriched for real interactions rather than background artifacts’, because I don't think that the miRDB target set analysis really tests anything about them being ‘between miRISCs’ (versus say simply free mRNA).

We thank the reviewer for the comment. We updated the corresponding sentence as you suggested (please see the highlighted sentence on page 10.)

Refs on pg 8 “we employed chimeric-eCLIP-seq^{27,28}” should just be ref #16; I'm not really sure how either of 27 or 28 are relevant here

We thank the reviewer for suggesting appropriate references. We updated the references in the text and the reference list.

I'd still like a bit more clarity in Sup Table 4 – I still don't totally understand what those numbers actually are (the

most straightforward interpretation is that it's the # of chimeric peaks identified at that gene divided by the length in nt? But that seems like a very strange metric – it seems like just the # of chimeric peaks would be more useful to provide)

The numbers represent the anticipated reads for the peak, allowing us to apply a Poisson model for peak region identification, as demonstrated in the CLIPper study (PMID: 24213538, cited in the initial submission). It also serves as a quantification metric.

REVIEWERS' COMMENTS

Reviewer #3 (Remarks to the Author):

The authors have addressed my concerns

REVIEWERS' COMMENTS

Reviewer #3 (Remarks to the Author):

The authors have addressed my concerns

We would like to express our sincere gratitude to the reviewer for the constructive comments and suggestions throughout the revisions.